# Intrinsic p53 activation restricts gammaherpesvirus driven germinal center B cell expansion during latency establishment

Shana M. Owens [1,5], Jeffrey M. Sifford[1,5], Gang Li[1], Steven J. Murdock[1], Eduardo Salinas [1], Darby Oldenburg [2], Debopam Ghosh[1], Jason S. Stumhofer [1], Intawat Nookaew [3], Mark Manzano [4] & J. Craig Forrest [4] ✉

Gammaherpesviruses are DNA tumor viruses that establish lifelong latent infections in lymphocytes. For viruses such as Epstein-Barr virus and murine gammaherpesvirus 68, this is accomplished through a viral gene-expression program that promotes cellular proliferation and differentiation, especially of germinal center B cells. Intrinsic host mechanisms that control virus-driven cellular expansion are incompletely defined. Using a small-animal model of gammaherpesvirus pathogenesis, we demonstrate in vivo that the tumor suppressor p53 is activated specifically in B cells latently infected by murine gammaherpesvirus 68. In the absence of p53, the early expansion of murine gammaherpesvirus 68 latency greatly increases, especially in germinal center B cells, a cell type whose proliferation is conversely restricted by p53. We identify the B cell-specific latency gene M2, a viral promoter of germinal center B cell differentiation, as a viral protein sufficient to elicit a p53-dependent anti-proliferative response caused by Src-family kinase activation. We further demonstrate that Epstein-Barr virus-encoded latent membrane protein 1 similarly triggers a p53 response in primary B cells. Our data highlight a model in which gammaherpesvirus latency gene-expression programs that promote B cell proliferation and differentiation to facilitate viral colonization of the host trigger aberrant cellular proliferation that is controlled by p53.

Gammaherpesviruses (GHVs) are DNA tumor viruses that include the human pathogens Epstein-Barr virus (EBV) and Kaposi sarcoma-associated herpesvirus (KSHV) and the rodent pathogen murine gammaherpesvirus 68 (MHV68), among others. Following a period of productive viral replication in various host tissues, GHVs establish lifelong chronic infections[1,2]. This phase of the infectious cycle, referred to as latency, is maintained in B cells. The lifelong infections caused by GHVs place the host at risk for numerous cancers, including Burkitt lymphoma, Hodgkin lymphoma, primary effusion lymphoma, multicentric Castleman disease, and many others[3,4]. Moreover, GHV-related malignancies represent serious complications for AIDS patients and transplant recipients[5]. However, the precise mechanisms by which GHVs promote oncogenesis are incompletely defined, as are the host pathways that prevent cellular transformation due to GHV infection.

[1]Dept. of Microbiology and Immunology and Center for Microbial Pathogenesis and Host Inflammatory Responses, University of Arkansas for Medical Sciences, Little Rock, AR, USA. [2]Gunderson Research Institute, LaCrosse, WI, USA. [3]Dept. of Biomedical Informatics, University of Arkansas for Medical Sciences, Little Rock, AR, USA. [4]Dept. of Microbiology and Immunology, Center for Microbial Pathogenesis and Host Inflammatory Responses, and Winthrop P. Rockefeller Cancer Institute, University of Arkansas for Medical Sciences, Little Rock, AR, USA. [5]These authors contributed equally: Shana M. Owens, Jeffrey M. Sifford. ✉e-mail: JCForrest@uams.edu

To establish and maintain latency, GHVs stimulate and usurp normal B cell differentiation processes[6]. This is accomplished through the actions of viral noncoding RNAs and oncoproteins, which, upon infection of naïve B cells, lead to an activated phenotype reminiscent of a germinal center (GC) reaction[7,8]. GC B cells are highly proliferative, with an estimated doubling time of every 6 hours[9]. GC B cells also undergo mutagenic processes known as somatic hypermutation and class-switch recombination, which change the antigenic specificity and effector functions of the B cell receptor and secreted antibodies[10]. The GC reaction ultimately results in the emergence of long-lived memory B cells or antibody-secreting plasma cells that respectively enable recall responses upon reinfection and facilitate elimination of pathogens[11]. It is hypothesized that stimulating GC reactions enables GHV-infected B cells to increase in number dramatically as the cells rapidly proliferate while also allowing a mechanism for the establishment of long-term latency in the circulating memory B cell pool[12]. This model of GHV latency establishment is largely based on EBV infection and immortalization of primary B cells in culture, but it is also supported by kinetic evaluations of cell types that harbor MHV68 in vivo and through analyses of mutant viruses following experimental infection of mice [reviewed in refs. 13,14].

For instance, the MHV68 homolog of the latency-associated nuclear antigen (mLANA) encoded by ORF73 is essential for latency establishment after intranasal inoculation of mice[15–17]. In addition to mediating maintenance of the viral episome during latent cell division, mLANA also promotes c-Myc stabilization, which likely influences latent cell survival[18]. The ORF72-encoded viral cyclin D ortholog (v-Cyclin) is a bona fide oncogene[19], and it is important for viral replication and reactivation from latency[20,21]. A viral Bcl-2 homolog (v-Bcl-2) encoded by the M11 open-reading frame, prevents activation-induced cell death to support viral latency[22–24]. The M2-derived gene product is a scaffold protein that in and of itself promotes B cell differentiation by activating Src family kinases, phospholipase C-gamma (PLC), and downstream transcription via nuclear factor of activated T cells (NFAT) and interferon regulatory factor 4 (IRF4)[25–28]. Interestingly, M2's role in facilitating MHV68 latency establishment appears to function almost exclusively in the GC B cell compartment[29,30]. EBV likewise uses a coordinated gene expression program that facilitates latency establishment and promotes cellular proliferation and differentiation[1]. Of note, latent membrane protein 1 (LMP1) mimics the activating and pro-survival signaling functions of CD40[31–33], a B cell surface receptor that interacts with CD40L on T follicular helper cells to facilitate GC reactions[10]. M2 and LMP1 are hypothesized to be functional homologs despite no discernable sequence homology[14].

While many of the viral oncogenes that promote cellular proliferation, survival, and differentiation have been studied extensively, less is known regarding the potential for cellular tumor suppressor pathways to be activated in response to manipulation of host-cell physiology by GHV proteins during latency establishment. It is noteworthy that EBV, KSHV, and MHV68 each encode viral proteins capable of inducing cell-cycle entry and transformation[19,32,34–36], yet none of these viruses efficiently transforms infected cells. This suggests that the viral latency-associated gene expression programs induce host cell responses that limit the potential for cellular transformation during infection. For example, although most, if not all cells are infected and begin to proliferate after EBV infection of primary B cells in culture, only a small percentage of infected cells immortalize to become lymphoblastoid cell lines (LCLs)[37–40]. This is because EBV-induced cellular proliferation triggers the activation of ataxia telangiectasia mutated (ATM) and checkpoint kinase 2 (Chk2), DNA damage response (DDR) kinases that inhibit cellular proliferation and subsequent immortalization[37]. KSHV also triggers ATM activation and proliferative arrest upon infection of primary endothelial cells[41]. Like EBV, MHV68 can immortalize primary cells in culture[42]. Although host restriction factors are not yet known, the process is similarly inefficient, requiring

weeks of culture prior to the outgrowth of transformed cells. While DDR pathways appear to restrict GHV-mediated cellular transformation in culture, tumor suppressor responses that restrict GHV infection and immortalization of cells in vivo during natural infection of a host are poorly defined and may differ from responses occurring in cell culture. For instance, a role for ATM in restricting MHV68 infection in vivo was tested; however, ATM deficiency did not promote enhanced latent MHV68 infection or cancer[43]. Rather, ATM facilitates B cell latency[44,45], suggesting that ATM does not impose a critical barrier to latent MHV68 infection or related oncogenesis in vivo.

Another candidate host protein for intrinsic latency restriction is tumor suppressor p53. p53 is considered a master regulator of genetic integrity, a postulate emphasized by the presence of inactivating p53 mutations in approximately half of all human cancers[46]. Though constitutively expressed, p53, under normal physiologic cellular conditions, is constantly targeted for proteolytic degradation by proteins such as ubiquitin ligase mouse double-minute 2 (MDM2)[47]. In response to potentially genotoxic cellular stresses, including DNA damage, oncogene expression, and viral infection, p53 becomes stabilized and activated through a variety of post-translational modifications[47]. Active p53 regulates transcription of numerous cellular genes involved in cell-cycle arrest, DNA repair, and apoptosis[48]. Interestingly, p53 is stabilized, and p53-responsive transcripts are induced during B cell immortalization by EBV[49], and the KSHV-encoded cyclin D ortholog is sufficient in and of itself to induce p53[34,41,50]. Exogenous induction of p53 by MDM2 inhibition reduces LCL formation by EBV[51], and conversely, p53 inhibition facilitates endothelial cell proliferation following KSHV infection[41]. Although both EBV and KSHV encode latency proteins demonstrated to inhibit p53 in various biochemical evaluations[52–54], established cell lines in which these viral inhibitors of p53 are highly expressed remain responsive to p53 agonists[49,55], that these proteins inefficiently suppress p53 functions in latently infected cells. Moreover, it is somewhat counterintuitive that a virus that establishes a life-long chronic infection would universally inhibit a tumor suppressor as critically important as p53. However, p53 mutations do occur frequently in endemic Burkitt lymphoma (eBL), a cancer characterized by stable EBV infection and the presence of chromosomal translocations, which suggests that p53 functions limit GHV-related cancers in vivo[2]. Whether p53 is activated to limit GHV infection and tumorigenic potential has not been directly tested using in vivo models of pathogenesis.

Here, we describe experiments that test the hypothesis that p53 is activated in response to GHV manipulation of B cell physiology during latency establishment. Using MHV68 infection of WT and p53-deficient mice, we define the functions of p53 in controlling viral latency establishment and maintenance in vivo. We employed a targeted sufficiency screen in primary B cells to identify specific viral gene products from MHV68 and EBV that are latency-related activators of p53, and we confirmed the MHV68 p53 agonist by comparing mutant and WT MHV68 in vivo. Finally, we performed mutational and pharmacologic inhibitor analyses that highlight key signaling events by which the viral factors promote p53 activation. Through this work, we demonstrate that virus-mediated manipulation of B cell activation and differentiation is countered by an intrinsic host tumor suppressor response to limit early proliferative events during latency establishment by a DNA tumor virus.

## Results
### MHV68 induces p53 during latency establishment
Tumor suppressor p53 is activated in response to aberrant cellular proliferation, differentiation, and genotoxic stress, especially when induced by cellular and viral oncogenes[1,34,56]. In order to establish latency, GHVs encode proteins that promote cellular proliferation and differentiation[3,4,13], which could hypothetically be recognized by the p53 pathway as potentially oncogenic events. While it is known that

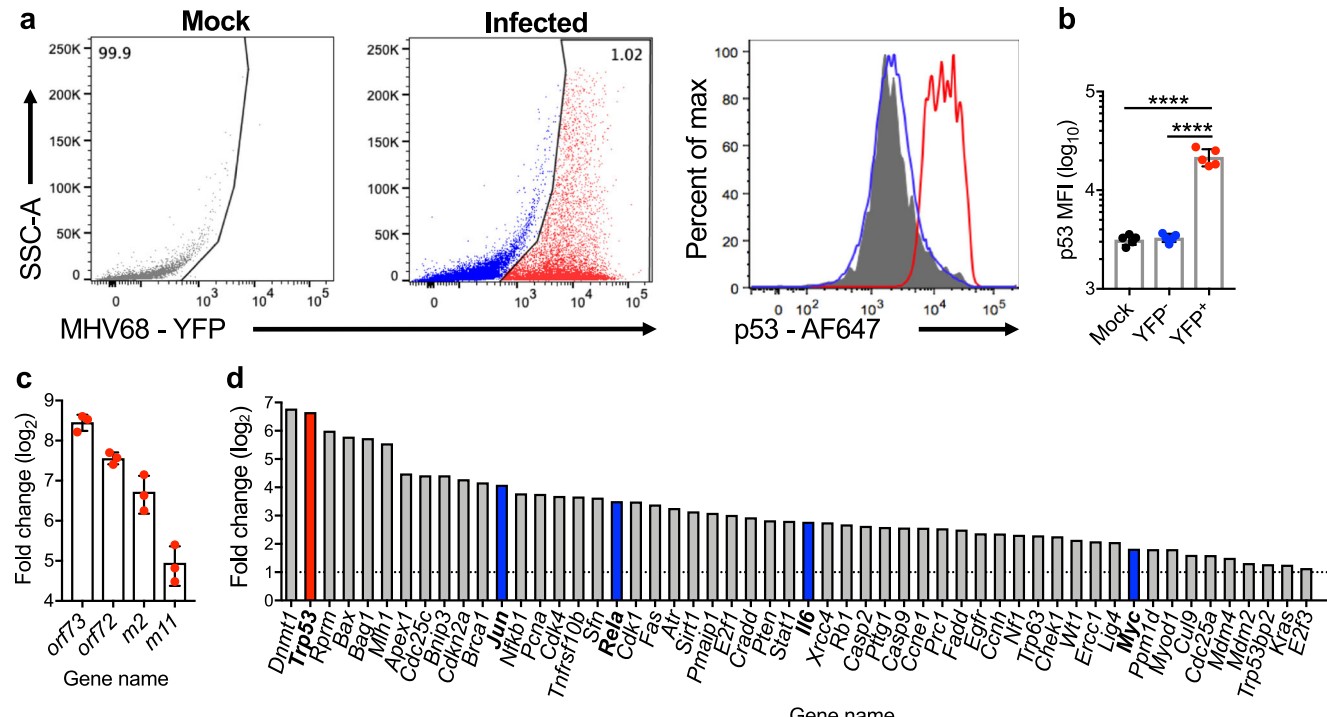

**Fig. 1 | p53 is induced and active in latently infected splenocytes. a** Flow cytometry analysis of p53 induction in infected (YFP$^+$, red) and uninfected (YFP$^-$, blue) spleen cells on day 16 after mock-infection (gray) or infection of C57BL/6 mice with $10^4$ PFU of H2B-YFP MHV68. **b** Quantification of data in (**a**) represented as mean +/− SEM. One-way ANOVA $p < 0.0001$. Each dot represents an individual mouse. 2-3 independent infections with a minimum of 3 mice per group. **c**, **d** Quantitative RT-PCR analyses comparing MHV68 infected to uninfected splenocytes after sorting on day 16 post-infection. The virus-dependent genetic marking strategy to identify infected cells and confirmation of infection is shown in Supplementary Fig. 2. Viral latency transcript abundance is shown in (**c**). Fold change represented as mean +/− SD. Profiler p53 pathway array data are shown in (**d**). Relative transcript abundance was determined by comparing virus-positive to virus-negative cells after sorting using the ΔΔC$_T$ method. The trp53 transcript is highlighted in red. Transcripts previously evaluated in MHV68 infection are denoted with blue bars.

p53 is induced by GHV infection of many cell types in culture[41,57], its activity is putatively inhibited by certain latency gene products[1,56]. However, whether p53 is activated and functional during GHV latency establishment in a living host is not known.

To begin to define the relationship between p53 and GHV latency, we infected mice with an MHV68 recombinant virus that encodes YFP to enable direct identification of infected cells by flow cytometry[58] and evaluated p53 induction in spleens of infected mice on day 16 after infection. At this time point, acute viral replication has waned, latent viral burdens in the spleen are at their peak, and the majority of infected B cells are proliferating and exhibit a germinal center phenotype[58–60]. In this analysis, p53 levels were significantly higher specifically in the MHV68-infected cells (YFP$^+$) than in mock-infected animals or the uninfected cells (YFP$^-$) in spleens of infected mice (Fig. 1a, b). As a downstream transcriptional target of the type I-interferon (IFN-I) signaling pathway during viral infections[61], we reasoned it possible that p53 was simply activated as part of the innate antiviral response to MHV68 infection; to determine if p53 activation during MHV68 latency was due to IFN-I signaling, we infected mice lacking the type I-IFN receptor (IFNAR1$^{-/-}$). Again, on day 16 post-infection p53 induction was evident almost exclusively in YFP$^+$ splenocytes at levels that were equivalent in both IFNAR$^{+/+}$ and IFNAR1$^{-/-}$ mice (Supplementary Fig. 1). These data indicate that p53 activation during MHV68 latency establishment is independent of IFN-I signaling and occurs specifically within infected cells.

Since p53 primarily functions as a transcription factor once stabilized, we next sought to evaluate p53-related transcription within latently infected cells. We infected inducible fluorescent reporter mice[62] with a recombinant MHV68 virus that expresses Cre recombinase to induce fluorescent protein expression[63,64] as a means to mark infected cells. After sorting fluorescent cells and confirming infection

by limiting-dilution PCR (LD-PCR)[65] and p53 induction by flow cytometry (Supplementary Fig. 2a, c), we performed p53 pathway RT-PCR array analyses comparing infected cells, which were approximately 96% CD19$^+$/B220$^+$ B cells (Supplementary Fig. 2b), to uninfected B cells from infected animals. Further confirming that the cells were indeed infected, viral latency transcripts were highly enriched in the fluorescent cell population, as were numerous p53-associated transcripts (Fig. 1c, d). Of note, the p53 transcript *Trp53* was ca. 100-fold higher in infected cells compared to uninfected cells, as were other transcripts with previously demonstrated roles in MHV68 latency, such as *Myc*, *Il6*, and *Nfkb1*[18,66,67]. Together these data support the conclusion that p53 is stabilized and transcriptionally active in vivo during MHV68 latency establishment. The observation that p53 induction occurred almost exclusively in infected cells using two independent marking strategies, as well as IFNAR1$^{-/-}$ mice, demonstrates that p53 activation is not a general consequence of viral infection as might be expected of a typical antiviral response, but rather represents a cell-intrinsic reaction to MHV68 infection.

**MHV68 latency establishment is enhanced in p53-deficient mice**
Given that p53 induction was specific to infected cells, we hypothesized that p53 functions in an intrinsic cellular response that controls MHV68 latency establishment. If correct, we predicted that the absence of p53 would correlate with enhanced MHV68 latency establishment. To test this hypothesis, we evaluated MHV68 infection of both p53$^{+/+}$ and p53$^{-/-}$ mice in side-by-side comparisons. Remarkably, on day 16 post-infection nearly 8% of cells in spleens of p53$^{-/-}$ animals were YFP$^+$ compared to 0.6% of cells in WT mice (Fig. 2a). In agreement with YFP$^+$ percentages, the frequency of splenocytes harboring MHV68 genomes was 14-fold higher in p53$^{-/-}$ animals than WT littermates, with approximately 1 in 20 p53$^{-/-}$ cells infected vs. 1 in ca. 280 WT cells

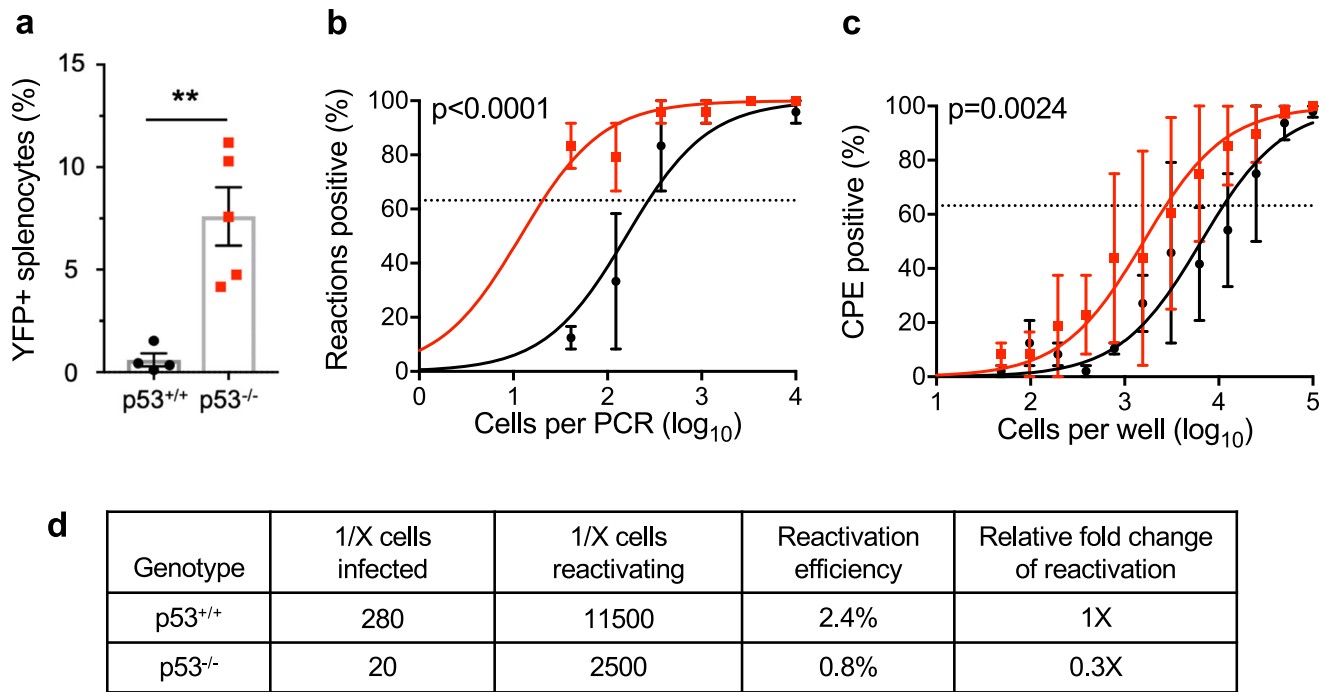

**d**

| Genotype | 1/X cells infected | 1/X cells reactivating | Reactivation efficiency | Relative fold change of reactivation |
|---|---|---|---|---|
| p53[+/+] | 280 | 11500 | 2.4% | 1X |
| p53[-/-] | 20 | 2500 | 0.8% | 0.3X |

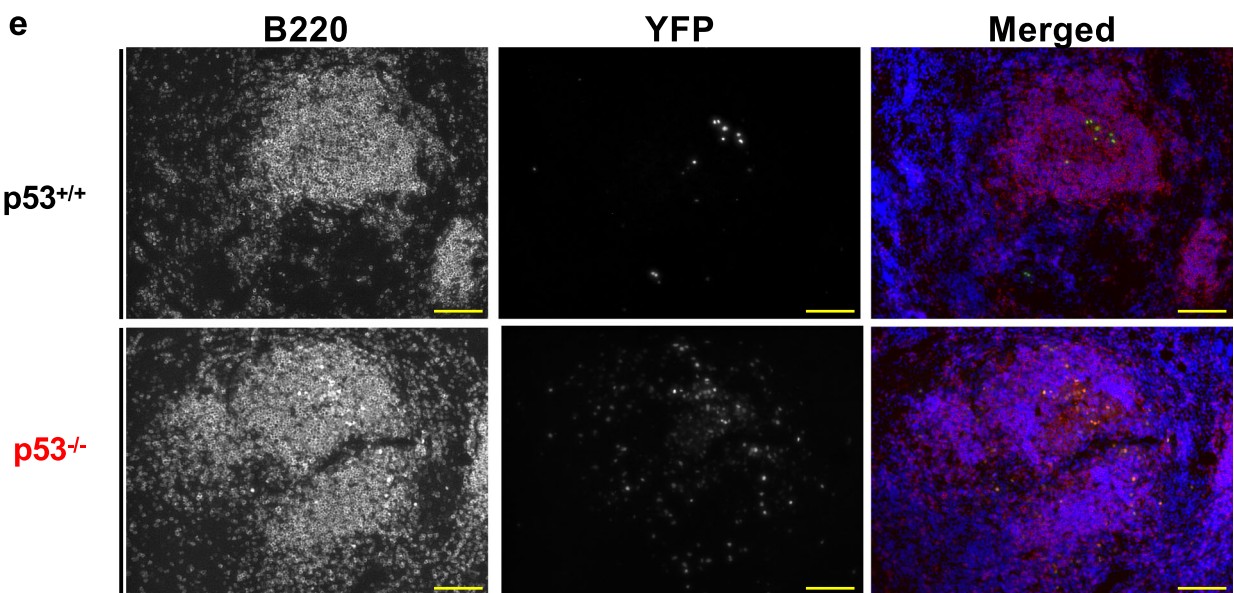

**Fig. 2 | MHV68 latency establishment is enhanced in p53-deficient mice. a** Flow cytometry analyses to determine the percentage of infected splenocytes (YFP⁺) for each mouse genotype at 16 days after mock infection or infection with $10^4$ PFU of H2B-YFP MHV68. Data represent means +/− SEM for 2-3 independent infections with a minimum of 3 mice per group. Each dot represents an individual mouse, n = 4-5. Welch's two-sided $t$ test, **$p < 0.005$. **b** Quantification of latently infected splenocyte frequencies by limiting-dilution PCR on day 16 post-infection. p53[-/-] or p53[+/+] mice were infected intranasally with $10^4$ PFU of H2B-YFP MHV68. Data represent means +/− SEM for 2 independent infections with a minimum of 3 mice per group. Extra sum-of-squares F test $p < 0.0001$, $F = 40.36$ (1, 22). **c** Quantification

of latent MHV68 reactivation efficiency from infected p53[-/-] or p53[+/+] splenocytes on day 16 post-infection. Reactivation was detected by scoring cytopathic effect in a limiting-dilution ex vivo culture. Data represent means +/− SEM for 2-3 independent infections with a minimum of 3 mice per group. Extra sum-of-squares F test $p = 0.0024$, $F = 10.36$ (1, 46). Replicate graphs in Supplementary Fig. 3. **d** table of values from (**a**, and **c**). **e**, Immunohistochemical visualization of B cells (B220⁺) and MHV68 infected cells (YFP⁺) in spleen sections from p53[-/-] or p53[+/+] mice on day 16 after infection. Representative images of splenic follicles are shown. Scale bars indicate 100 µm. Additional representative images in Supplementary Fig. 4.

(Fig. 2b, d). Moreover, immunofluorescence analyses of spleen sections revealed that infected cells were more readily detected in B220⁺ B cell follicles of p53-null mice in comparison to WT mice (Fig. 2e and Supplementary Fig. 4).

Previous studies suggest that p53 plays roles in innate and adaptive immune responses to certain infections[68–70], so we reasoned mice

lacking p53 might be less able to control MHV68 infection. In evaluating acute viral replication, we found no difference in MHV68 infection of WT and p53 knockout mouse lungs on day 7 post-infection (Supplementary Fig. 3a), the typical peak of acute viral replication in this tissue[71]. This result suggests that p53 is not critical for controlling the acute phase of MHV68 infection in mice. While the frequency of

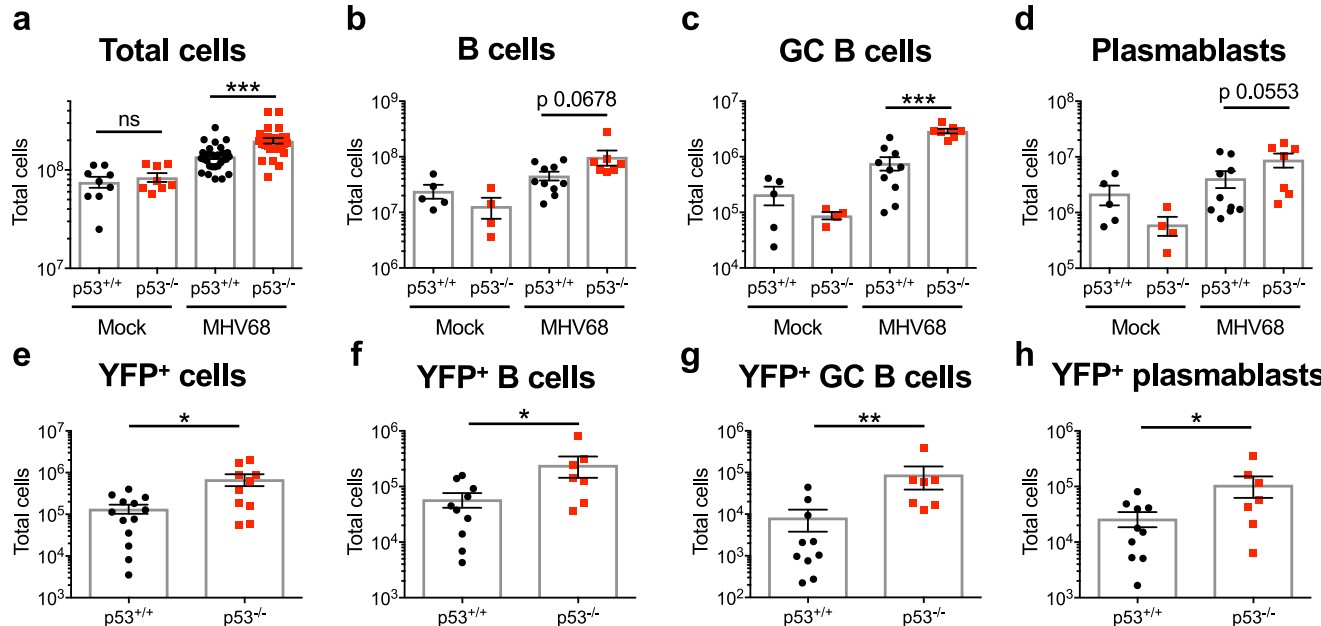

**Fig. 3 | B cell expansion and infection is increased in p53-deficient mice.**
**a–d** Quantification of total splenocytes, B cells, GC B cells, and plasmablasts 16 days after mock infection or infection of p53[+/+] or p53[-/-] mice with 10[4] PFU of H2B-YFP MHV68. Flow cytometry was performed to identify specific B cell populations. B cells were gated as CD19[+]/B220[+] in (**b**) GC cells as GL7[+]/CD38[lo] sub-population of CD19[+]/B220[+] B cells in (**c**) and plasmablasts as CD138[+]/B220[lo] in (**d**). Gating strategies are shown in Supplementary Fig. 9. Other lymphocyte populations were also quantified and were equivalent in p53[-/-] and p53[+/+] spleens (Supplementary Fig. 6). **e–h** Quantification of infected (YFP[+]) total splenocytes, B cells, GC B cells, and plasmablasts as identified in (**a–d**) on day 16 after infection. **a–h** Data represent means +/− SEM for 2-3 independent infections with a minimum of 3 mice per group. Each dot represents an individual mouse. Mann-Whitney unpaired two-sided *t* test, *$p < 0.05$, **$p < 0.005$, ***$p < 0.0005$, ****$p < 0.0001$.

latently infected splenocytes with the capacity to reactivate upon ex vivo culture was enhanced modestly compared to WT mice (Fig. 2c and Supplementary Fig. 3b), the increase correlated with the higher frequency of cells harboring MHV68 in p53[-/-] mice (Fig. 2d), and we did not detect evidence of ongoing persistent viral replication[72,73]. The absence of increased acute replication, enhanced reactivation, or persistent replication, outcomes largely controlled by innate immune responses[74–78], suggests that p53 is not directly involved in these host defense mechanisms. Finally, MHV68-specific adaptive immune responses also developed normally in p53[-/-] mice, as virus-specific IgG in serum, T cell activation, and induction of MHV68 antigen-specific CD8[+] T cells were comparable in both WT and p53-null animals infected with MHV68 during latency establishment (day 16) and maintenance (day 42) as immune activation wanes (Supplementary Figs. 5 and 6). Together, these data demonstrate that the absence of p53 correlates with enhanced MHV68 latency establishment and supports the hypothesis that p53 mediates intrinsic cellular resistance to chronic MHV68 infection.

**p53 controls cellular proliferation during MHV68 latency establishment**
As mentioned above, MHV68 and other GHVs encode latency-associated gene products that promote the expansion of B cells that serve as reservoirs for latent infection. For MHV68 this results in a syndrome akin to infectious mononucleosis caused by EBV[79]. In addition to enhanced latent MHV68 infection in p53-deficient animals, it was readily apparent that there was also a significantly greater infection-related increase in the number of cells per spleen compared to WT mice (Fig. 3a). This observation suggested that p53 acts to control cellular expansion induced by the virus. In support of this hypothesis, quantification of specific lymphocyte populations by flow cytometry revealed enhanced expansion of B cells in p53[-/-] mice (Fig. 3b), while T cell and natural killer (NK) cell numbers were similar between WT and knockout animals (Supplementary Fig. 6). For

particular B cell subsets, the number of GC B cells was significantly increased (Fig. 3c), but plasmablasts were only modestly affected by the absence of p53 (Fig. 3d). Reflecting MHV68's latent tropism for B cells and usage of the GC B cell compartment during latency establishment, the number of infected B cells was significantly increased in p53 knockout mice, with the MHV68 infected subset of GC B cells exhibiting a ca. 10-fold relative increase (Fig. 3e–h). Infection of plasmablasts also was significantly enhanced, which may result from the differentiation of infected GC B cells. In contrast to infection of p53[-/-] mice with WT virus, infection with a latency-deficient mutant virus lacking the MHV68 homolog of the latency-associated nuclear antigen (mLANA-null) did not cause increased expansion of splenocyte populations (Supplementary Fig. 7). Since this mutant virus can still undergo acute replication and stimulate adaptive immunity[15,16], these data suggest that the enhanced splenocyte expansion that occurs in p53[-/-] mice requires the presence of latent MHV68 and is not simply a consequence of the general immune response to viral infection.

Given that GHVs drive B cell proliferation in order to establish and maintain latency[12,13] and p53 restricts cellular proliferation or promotes apoptosis downstream of oncogene expression[56,80–82], we hypothesized that p53 functions to limit MHV68-driven cellular proliferation and/or induce death in infected cells. Using in vivo 5-ethynyl-2'-deoxyuridine (EdU)-incorporation to label cells actively replicating their DNA, we detected similar basal levels of EdU incorporation in both p53[-/-] and WT animals that were mock-infected. Infection with MHV68 led to an increase in the number of EdU[+] GC B cells irrespective of genotype; however, spleens of p53-deficient mice contained twice as many EdU[+] GC B cells than WT mice (Fig. 4a). The number of apoptotic cells after infection was only modestly reduced in the absence of p53 as determined by flow cytometry-based active caspase 3/7 staining (Fig. 4b). These data suggest that a mechanism by which p53 limits MHV68 latency establishment is through restriction of cellular proliferation, an interpretation supported by the enhanced expansion and infection of the highly proliferative GC B cell compartment in p53[-/-] mice.

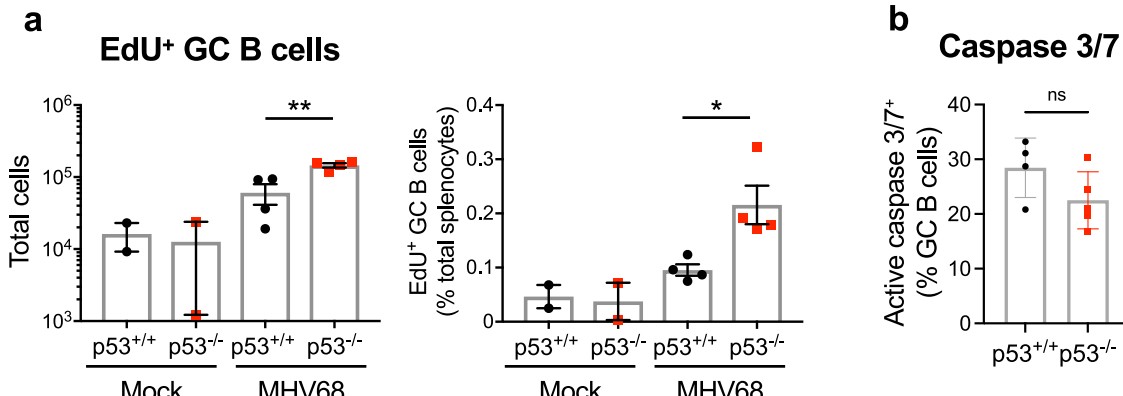

**Fig. 4 | p53 controls GC B cell proliferation during MHV68 latency establishment. a** Quantification of GC B cells undergoing DNA replication on day 16 after mock infection or infection of p53-/- and p53+/+ mice with MHV68. EdU was injected intraperitoneally 2 h prior to sacrifice to label newly replicated DNA. Click chemistry was performed and EdU + GC B cells (gated as CD19+/B220+/GL7+/CD38lo) were detected by flow cytometry. Total (left panel) and percent EdU + GC B cells per spleen (right panel) are shown. Each dot represents an individual mouse, n = 4.

**b** Flow cytometric quantification of GC B cells (gated as CD19+B220+GL7hiCD38lo) undergoing apoptosis on day 16 after infection of p53-/- and p53+/+ mice with MHV68. Apoptotic cells were identified as active caspase 3/7+ after ex vivo staining. Gating strategies are shown in Supplementary Fig. 10. Data represent means +/− SEM. Mann-Whitney unpaired two-sided t test, * p < 0.05, **p < 0.005. Each dot represents an individual mouse, n = 4-5.

## Long-term MHV68 latency is not enhanced in p53-deficient mice

After the initial virus-driven B cell activation and expansion associated with the establishment of MHV68 latent infection, both total cell numbers and the frequency of cells harboring viral genomes contract to levels that are maintained at relatively stable levels during long-term infection[59,60,83]. In testing the impact of p53 on long-term MHV68 latency, we found that viral genome-positive cells in spleens of p53-/- mice were ca. 5-fold lower than WT mice on day 42 post-infection, a phenotype that was maintained up to 100 days post-infection (Supplementary Fig. 8a, b). The total number of cells, including total B cells and GC and plasmablast subsets, were equivalent in spleens of both knockout and WT animals by day 42 after infection (Supplementary Fig. 8c–f). These data suggest that p53 is mainly activated to restrict virus-driven B cell expansion during latency establishment and does not play a critical role in controlling infection as latency is maintained over time. Moreover, it is notable that despite restricting initial latent colonization of the host, the modest reduction in latent viral burdens observed at later time points suggests that p53 induction during latency establishment has a pro-viral role in long-term latent infection by MHV68.

## The M2 latency protein promotes p53 activation in primary B cells

Since p53 was stabilized specifically in MHV68 infected cells where it appeared to restrict GC B cell expansion during early latent infection, we hypothesized that viral proteins involved in latency establishment are responsible for activating p53 in B cells. To test this hypothesis, we performed a screen in which primary B cells were transduced with retroviruses that encode the major MHV68 latency genes, *M2*, *M11*/v-BCL2, *ORF72*/v-Cyclin, and *ORF73*/mLANA[15,16,22,84,85] and evaluated p53 induction by flow cytometry. Frameshift-stop mutant versions of each gene were used as negative controls, and high-level transcription of each construct was verified by quantitative RT-PCR (Supplementary Fig. 11a–c). In this screen, only the M2-encoding retrovirus led to p53 induction (Fig. 5a). Similar to MHV68 infected splenocytes, increased p53 detection was only observed in M2 transduced cells and not bystander cells within the culture (Supplementary Fig. 11e), which further supports the notion that p53 induction is an intrinsic cellular response to viral gene expression and does not occur through paracrine mechanisms.

The M2 latency protein of MHV68 functions similarly to EBV latent membrane proteins to promote B cell proliferation and

differentiation[1,86]. M2 acts as a scaffold protein that activates Src-family kinases[28,87], triggering a signaling cascade that results in NFATc1 and IRF4-dependent production of the cytokine IL-10[26]. This signal transduction is mediated through a structural motif encompassing two tyrosine residues (Y120 and Y129)[25,27]. M2 function is specifically required in GC B cells to facilitate MHV68 latency establishment[30,88], and its expression in naive B cells results in a GC-like activation profile and BCR class switching that is dependent on IL-10[86]. Given that oncogenes are well known to activate cellular tumor suppressor responses[89], we reasoned that the M2-driven signal transduction pathway would be recognized as a potentially oncogenic stimulus by the cell, resulting in p53 activity. To test this hypothesis, we transduced primary B cells with control, WT M2-encoding retroviruses, and an M2 tyrosine mutant (Y120F/Y129F) to disrupt M2-dependent cellular activation. Unlike B cells transduced with retrovirus encoding WT M2, those transduced with M2 mutant (M2Tyr) retrovirus did not exhibit p53 induction (Fig. 5b), suggesting that residues in M2 required for signal transduction are responsible for activating p53 in primary B cells.

To further define the molecular pathways involved in M2-driven activation of p53, we transduced primary B cells with control or M2-expressing retrovirus in the presence or absence of specific pharmacologic inhibitors or an IL-10 inhibitory antibody using concentrations previously described to inhibit M2-related effects on B cells[26]. Remarkably, Src inhibitor PP2 completely blocked M2-mediated p53 induction while its inactive control analog PP3 had no effect (Fig. 5c). Cyclosporin A (CsA), an inhibitor of the calcineurin/NFAT pathway, partially prevented p53 induction, however M2-mediated p53 induction still occurred in the presence of an antibody that blocks the IL-10 receptor. While we cannot rule out potential off-target effects of the reagents tested, together the M2Tyr mutant and pharmacologic inhibitor data strongly suggest that p53 activation is primarily mediated by M2's activation of a Src-family kinase signal transduction cascade. This is also consistent with Src kinases acting upstream of NFAT and IL-10 upon expression of M2 in B cells, which agrees with previous studies[26].

Detecting p53 induction after expression of M2 in B cells seemed to contradict prior work suggesting that M2 promotes cellular proliferation and survival[26,86,90]. We, therefore, performed RNA-seq experiments as an unbiased means to evaluate transcription pathways regulated by M2 expression in B cells. In agreement with previous analyses[26], M2 transduction led to high-level transcription of *Il10* and *Irf4* (Fig. 5d). In comparisons of control to M2-transduced cells, gene set enrichment analysis (GSEA) revealed a significant positive

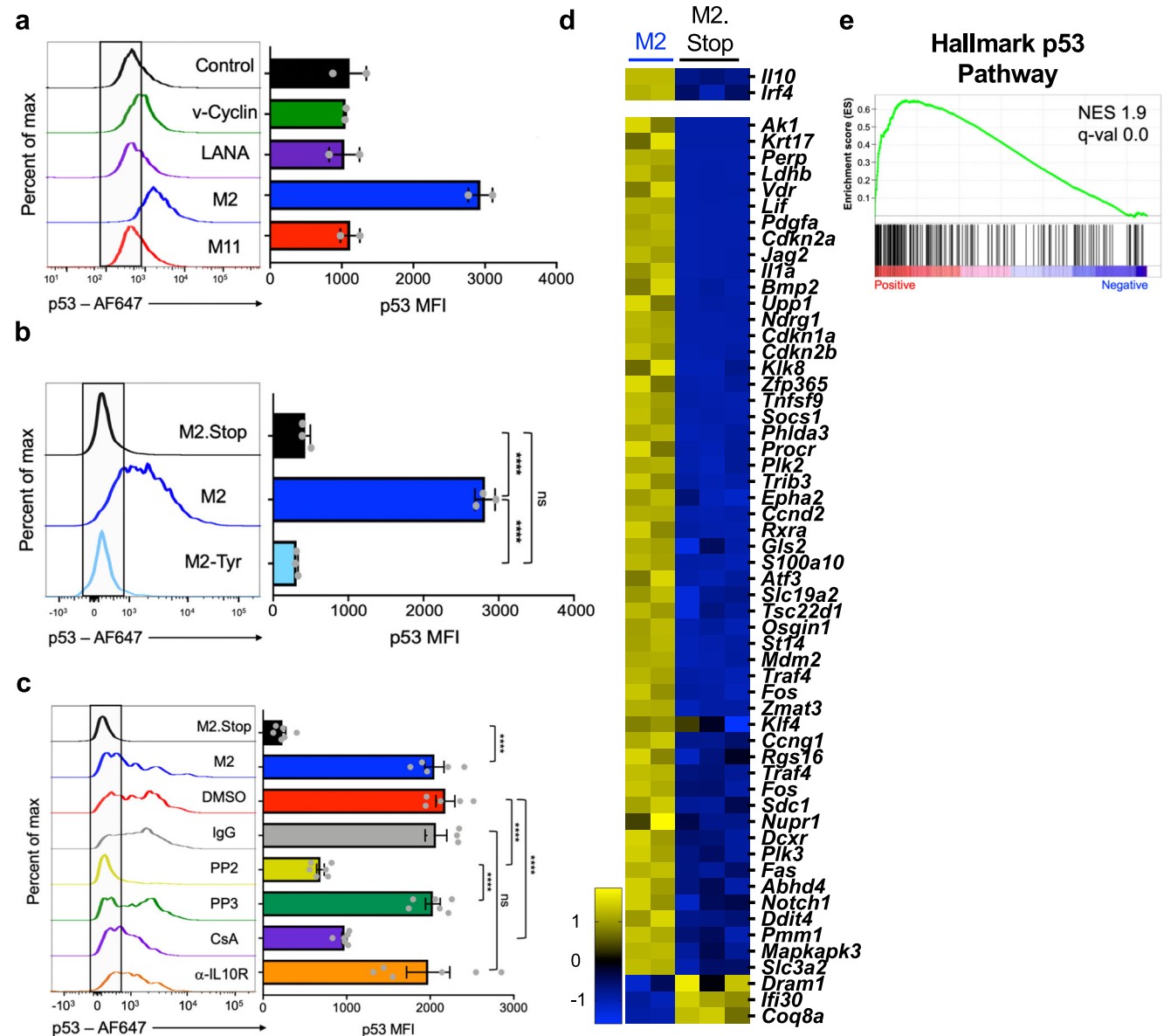

**Fig. 5 | MHV68 latency protein M2 causes p53 induction via Src-family kinase signaling. a, b** Flow cytometry analysis of p53 induction in primary murine B cells after transduction with retroviruses that encode MHV68 latency proteins. Representative histograms and mean fluorescence intensities of p53 staining in transduced cells are shown. Data in (**a**) show p53 staining 4 days after transduction of cells with retroviruses that encode v-Cyclin, LANA, M2, M11, or a frame-shift/translation-stop mutant M2 (M2.Stop) as a negative control. Data are biological replicates, $n = 2$. Data in (**b**) show p53 staining 2 days after transduction with M2.Stop, M2-WT, or a tyrosine mutant M2-Y120F/Y129F (M2-Tyr). Data represent mean ± SEM. Two-tailed Student's $t$ test, ns - not significant, *$p < 0.05$, and ****$p < 0.0001$. The experiment was performed in technical triplicate. Data in (**c**) show p53 staining 2 days after transduction with M2.Stop or M2-encoding retroviruses in the presence of DMSO (1:1000), mouse IgG (20 μg/mL), PP2 (10 μM), PP3 (10 μM), CsA (500 ng/mL), or rat anti-IL10R (20 μg/mL). Inhibitors were added 24 h post-transduction. Data represent mean ± SEM. Ordinary one-way ANOVA, ns - not significant and ****$p < 0.0001$. The experiment was performed in technical triplicate. Gating strategies and additional controls are shown in Supplementary Fig. 11. **d, e** RNA-seq was performed 4 days after transduction of primary B cells with M2 or M2.Stop retroviruses to evaluate changes in transcription. Data in (**e**) show gene set enrichment analysis (GSEA) and (**d**) hallmark gene z-score heatmaps comparing M2 to M2.Stop for the hallmark p53 gene set. Statistical scores are inset into the top right of analysis images. NES normalized enrichment score, q-val FDR-adjusted *p*-value. The genes presented were derived from GSEA and DEseq2 analysis of all genes with significant expression changes where the expression level increased from M2 to M2.Stop at least 1.5-fold and decrease from M2 to M2.Stop at least 1.5-fold. A table of hallmark gene sets with significant changes caused by M2 is shown in Supplementary Fig. 14.

enrichment of p53 hallmark transcripts upon M2 expression (Fig. 5d, e and Supplementary Fig. 14). These studies, therefore, demonstrate that M2 causes p53 induction and transcriptional activity in primary murine B cells.

## MHV68 M2 induces the p53 pathway in vivo
Having defined M2 as sufficient to promote p53 induction and transcription in primary B cells in culture, we next sought to determine

whether M2 promotes p53 activity in vivo. M2 functions primarily in GC B cells to facilitate latency establishment[88], which presented a complication since this was also the primary cell type impacted by the presence of p53 during MHV68 infection. We, therefore, utilized two specific reporters to facilitate comparisons between cells infected with either WT or M2-null MHV68. To mark GC B cells, we used AID-Cre Ai14 reporter mice, which express TdTomato upon expression of activation-induced cytidine deaminase (AID), a B cell-specific enzyme

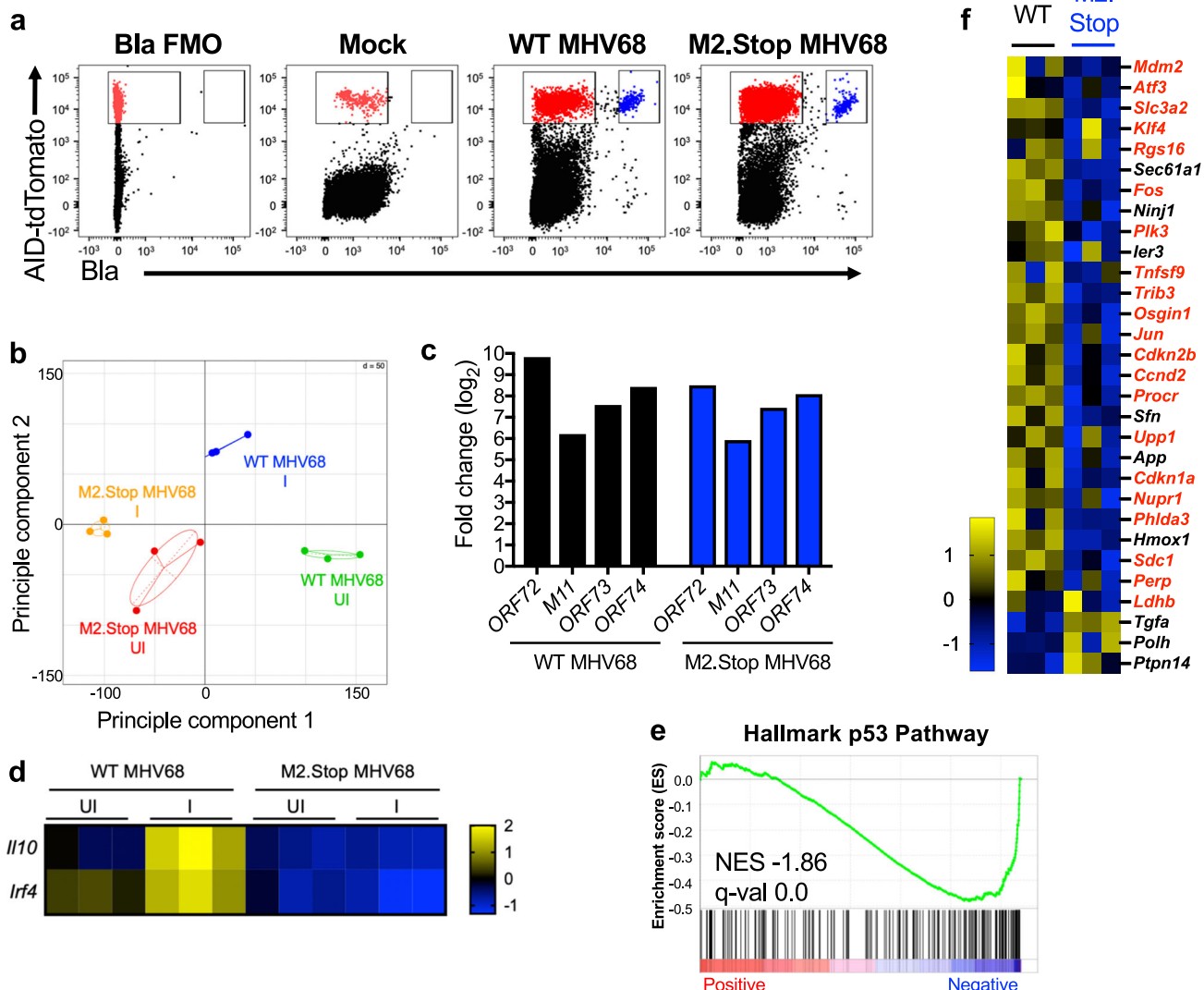

**Fig. 6 | M2 activates p53 pathway expression in vivo.** AID-cre Ai14 mice were infected intranasally with $10^4$ PFU of WT or M2.Stop 73.Bla MHV68. At 14 dpi infection, draining lymph nodes were isolated and treated with CCF4-AM. MHV68⁺ cells are defined by cleavage of CCF4 fluorescent substrate (Bla⁺). **a** Sorting strategy for AID⁺ 73.Bla MHV68 lymphocytes. The AID⁺ uninfected cell population marked in red and AID⁺ Bla⁺ cell population marked in blue. **b**–**f** Low input RNA-seq was performed on WT or M2.Stop MHV68 sorted cells to evaluate changes in transcription. **b** Principle component analysis (PCA) of low input RNA-seq data. **c** Viral latency transcript abundance compared to uninfected AID⁺ lymphocytes. **d** M2-associated transcript expression. Z-score heatmap comparing *Il10* and *Irf4*

expression in WT (UI and I) and M2.Stop (UI and I) populations. Data in (**e**) show gene set enrichment analysis (GSEA) comparing WT to M2.Stop MHV68 infection for the hallmark p53 gene set. Statistical scores are inset into the bottom left of analysis images. NES, normalized enrichment score. q-val, FDR-adjusted *p*-value. **f** Z-score heatmaps show average expression data for WT or M2.Stop MHV68 AID⁺ lymphocytes. The genes presented were derived from GSEA and DEseq2 analysis of all genes with significant expression changes where the expression level increased from WT to M2.Stop at least 1.5-fold and decrease from WT to M2.Stop at least 1.5-fold. M2-associated transcripts identified in Fig. 5 are denoted in red.

required for somatic hypermutation and class-switch recombination during GC reactions[62,91]. To identify infected cells, we utilized WT and M2.Stop MHV68 reporter viruses in which the enzyme beta-lactamase was fused to the episome-maintenance protein mLANA (73.Bla)[59], which is highly expressed in GC B cells[92]. This reporter was selected based on compatibility with TdTomato in flow cytometry analysis, the increased sensitivity to detect infected cells due to enzymatic amplification of fluorescent signal upon cleavage of the beta-lactamase substrate[59], and its compatibility with live-cell sorting for downstream RNA sequencing. The reporters used and the necessity for live-cell sorting, unfortunately, prevented a direct evaluation of p53 protein levels in these experiments. While M2 is essential for efficient viral dissemination and splenic latency establishment[93–95], M2-deficient viruses are detected in draining lymph nodes at levels comparable to WT MHV68 after IN infection[88]. We therefore infected AID-Cre Ai14

reporter mice with either WT 73.Bla or M2.Stop 73.Bla MHV68 and sorted infected AID⁺Bla⁺ and uninfected AID⁺Bla⁻ cells from mediastinal lymph nodes for ultra-low-input RNA-seq (Fig. 6a). Principle component analyses revealed that each specific group exhibited unique, non-overlapping transcription profiles that were similar between biological replicates within groups (Fig. 6b). Viral latency transcripts were highly enriched in Bla⁺ data sets of both M2.Stop and WT MHV68 (Fig. 6c), however, only AID⁺Bla⁺ samples from WT MHV68 infection showed increased transcription of *Il10* and *Irf4* (Fig. 6d), which is consistent with previous work demonstrating that M2 upregulates their expression[26,86]. Significant negative enrichment in Hallmark p53 Pathway GSEA comparisons between Bla⁺ M2.Stop and WT MHV68 infected B cells revealed that p53-related transcripts were generally lower in infected AID⁺ cells when M2 expression was ablated (Fig. 6e, f). The majority of differentially regulated p53-related transcripts in this

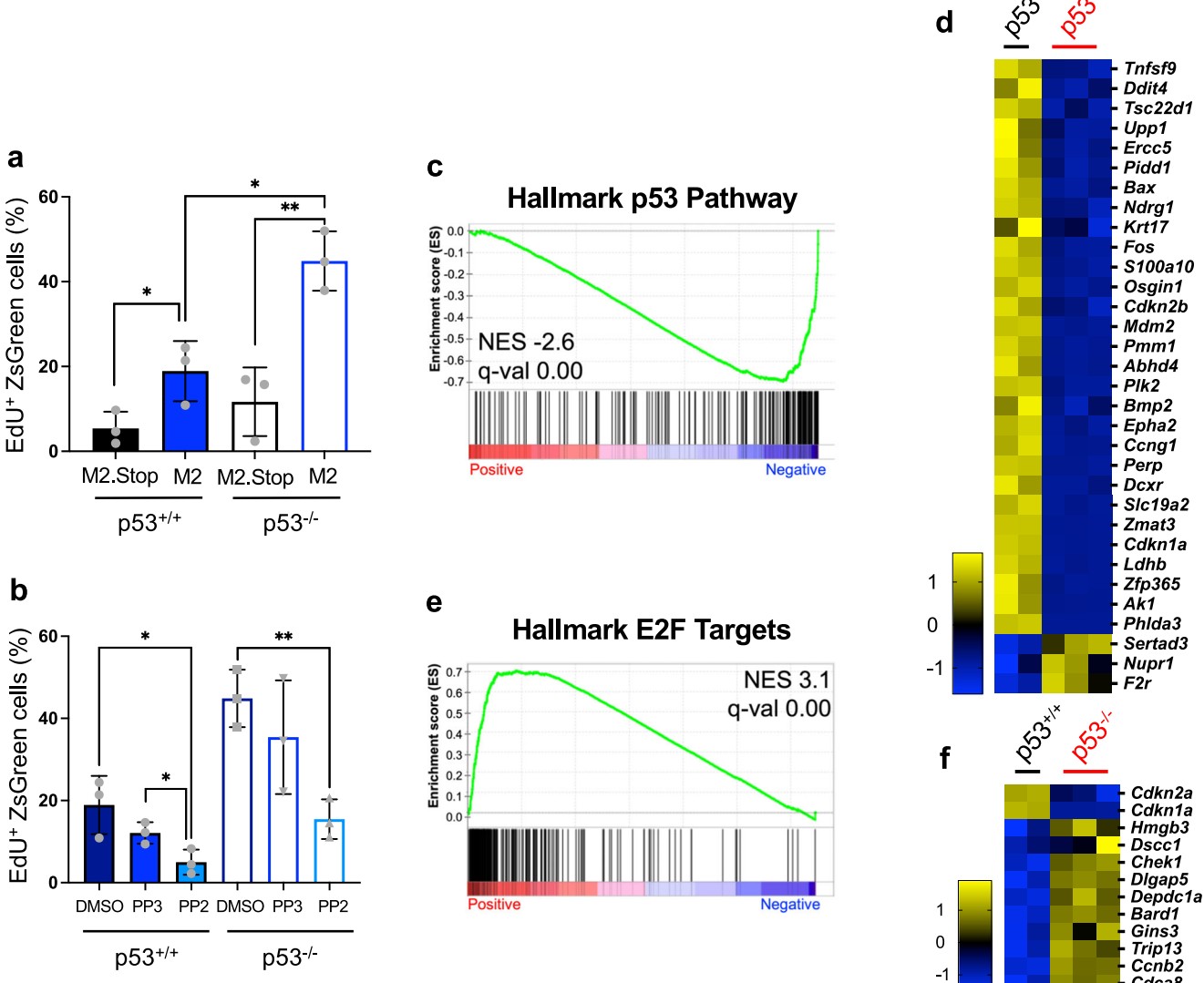

**Fig. 7 | p53 suppresses M2-driven B cell proliferation. a, b** Quantification of p53$^{-/-}$ or p53$^{+/+}$ B cells undergoing DNA replication after transduction with M2 or M2.Stop retroviruses. Cells were pulsed with 10 mM EdU for 4 h prior to harvest at 50 h post-transduction. Cells in (**b**) were treated with Src-family inhibitor PP2 (10 μM) or controls (DMSO at 1:1000, PP3 at 10 μM) 24 h post-transduction. EdU incorporation in newly synthesized DNA was labeled by Click chemistry. EdU$^+$ transduced B cells (gated on ZsGreen$^+$ cells) were detected by flow cytometry. Data represent means +/− SEM. Two-tailed Student's $t$ test, ns - not significant, *$p < 0.05$, ** $p < 0.01$. The experiment was performed in technical triplicate. Representative flow cytometry plots and additional controls are shown in Supplementary Fig. 12. **c−f** RNA-seq was performed 4 days after transduction of p53$^{-/-}$ or p53$^{+/+}$ B cells with M2 or M2.Stop retroviruses to evaluate changes in transcription. Data in (**c, d**) show gene set enrichment analysis (GSEA) and hallmark gene z-score heatmaps comparing M2 to M2.Stop for the hallmark p53 gene set. Data in (**e, f**) show the hallmark E2F gene set. Statistical scores are inset into the top right of analysis images. NES, normalized enrichment score. q-val, FDR-adjusted $p$-value. **d, f** z-score heatmaps show average expression data for M2 or M2.Stop expressing primary B cells. The genes presented were derived from GSEA and DEseq2 analysis of all genes with significant expression changes where the expression level increased from M2 to M2.Stop at least 1.5-fold and decrease from M2 to M2.Stop at least 1.5-fold. A table of hallmark gene sets with significant changes caused by M2 are shown in Supplementary Fig. 14.

comparison were also induced by M2 expression alone in primary B cells (highlighted in red text). Together with the demonstration that M2 in and of itself has the capacity to induce p53 and downstream transcription, these results strongly suggest that M2 is both necessary and sufficient to promote p53 pathway activation in infected B cells.

**p53 controls M2-driven cellular proliferation**

Given p53's known roles in restricting cell-cycle progression, M2-specific p53 pathway induction both in culture and in vivo, and the apparent p53-related control of GC B cell replication during MHV68 latency establishment in mice, we reasoned that p53 was restricting M2-driven B cell proliferation. To test this hypothesis, we evaluated EdU incorporation after M2-encoding retroviral transduction of WT

and p53$^{-/-}$ primary B cells. Confirming our hypothesis, EdU incorporation assays showed a large increase in the percentage of p53$^{-/-}$ cells undergoing DNA replication compared to WT cells upon M2 expression (Fig. 7a). As in WT cells, EdU incorporation was potently reduced by Src kinase inhibitor PP2 in p53-deficient cells, which further highlights the importance of Src family kinases in M2-mediated B cell activation (Fig. 7b). Importantly, RNA-seq analyses revealed that p53 hallmark genes, which include numerous anti-proliferative transcripts, were negatively enriched in p53$^{-/-}$ B cells after M2 transduction (Fig. 7c, d and Supplementary Fig. 14), while target genes of the cell-cycle promoting E2F pathway were induced in the absence of p53 (Fig. 7e, f). M2-responding transcripts *Irf4* and *Il10* remained highly induced in p53$^{-/-}$ cells, suggesting that p53 does not directly influence

transcription via the Src/NFAT/calcineurin pathway activated by M2 (Supplementary Fig. 12). Together, these data provide evidence that M2 drives Src family kinase-related cellular proliferation that is restricted by p53 activation in primary B cells.

### EBV LMP1 causes p53 activation in murine primary B cells

Analogous to MHV68, EBV induces GC-like responses in infected B cells, which the virus usurps to facilitate latency establishment[6,8,96]. In particular, the EBV "growth program" (also known as Latency III) involves the coordinated expression of several latency factors that lead to B cell activation, proliferation, and eventual immortalization into lymphoblastoid cell lines (LCLs)[97,98]. Prior studies demonstrated that EBV infection of primary human B cells elicited p53 stabilization[49], and p53 agonists limit LCL formation upon EBV infection[51]. We, therefore, hypothesized that distinct latency factors involved in the EBV growth program may trigger p53 induction, and we focused our analysis on LMP1 due to its functional similarities to M2[1] and previous reports that LMP1 expression correlates with p53 stabilization[99,100]. Consistent with our hypothesis, transduction of primary murine B cells with retroviruses encoding LMP1 resulted in increased p53 detection by flow cytometry (Fig. 8a). In contrast, dominant-negative LMP1 (LMP1$^{AAAG}$) encoding four point-mutations in residues critical for activating NF-κB, STAT, and AP-1 pathways[101] did not activate p53 in transduced B cells (Fig. 8b), which suggests that LMP1-dependent signal transduction leads to p53 activation. We again performed RNA-seq analyses on transduced cells and found that p53-related transcripts were significantly enriched in LMP1-transduced cells relative to control transductions (Fig. 8d, e), as were c-Myc signature transcripts (Supplementary Fig. 14). Comparisons of RNA-seq data from p53 competent and knockout cells expressing LMP1 indicated that enrichment of both c-Myc and p53-related transcription in these cells is linked to p53 expression (Supplementary Figs. 13d, e and 14). Interestingly, both LMP1- and M2-transduced p53$^{-/-}$ B cells formed large clusters reminiscent of an activated, blasting phenotype (Fig. 8c) that proliferated and survived in culture for 2-3 weeks (Supplementary Fig. 13b), whereas transduced WT cells could not be maintained. These data demonstrate that EBV-encoded LMP1 elicits p53 activation in mouse B cells in a manner that is consistent with p53 restricting cellular proliferation and survival. This parallel observation for an EBV latency protein suggests that antagonism of the virus-driven GC response by p53 is a common feature of GHV latency establishment.

## Discussion

GHVs take advantage of B cell proliferation and differentiation to facilitate the establishment of life-long latent infection of their hosts. This is accomplished through a coordinated latency gene-expression program that augments and mimics natural developmental processes that guide B cell activation and differentiation as part of the adaptive immune response to infection. We demonstrate here, using an in vivo model of GHV infection that p53 is induced specifically in infected cells during the period in which a murine GHV, MHV68, manipulates the B cell compartment to colonize its host. Although p53 is thought to be repressed by Bcl-6 in GC B cells to facilitate rapid cellular proliferation and mutagenesis of the B cell receptor[102], the absence of p53 correlated with large increases in the number of GC B cells, their proliferation, and infection of these cells by MHV68. Further supported by the detection of p53-responsive transcripts within latently infected cells, these data strongly suggest that p53 remains responsive in the GC B cell compartment and is capable of restricting GHV-driven B cell expansion as the virus establishes latency. Since p53 stabilization occurs during EBV infection of primary human B cells[49], p53 agonists restrict their immortalization[51], and LMP1 induces p53, we propose that p53 is a common intrinsic inhibitor of GHV latency establishment in B cells.

As a master regulator of cell-stress responses, p53 activation during viral infection has been evaluated for numerous viruses, such as influenza virus, herpes simplex virus (HSV), human cytomegalovirus (HCMV), HIV, SV40, adenovirus, and others, including MHV68 and EBV[49,81,103–105]. In fact, p53 was initially identified through its interaction with the SV40 oncoprotein, Large T antigen (T$_{ag}$)[106]. It is important to emphasize that these studies primarily focused on p53 function and inhibition during the productive replication phase of the viral infection cycle. For instance, p53 is potently inhibited by Large T$_{ag}$, adenovirus E1B-55K/E4orf6, and MHV68 mLANA and muSOX during lytic replication[104,107]. Identical productive viral replication in both WT and p53$^{-/-}$ mice highlights the potent inhibition of p53 during acute MHV68 replication. In contrast, p53 is an interferon-stimulated gene that participates in the antiviral response and host survival during HSV-1 infection of mice[108], whereas HCMV integrates p53 activity to promote lytic gene expression and viral replication[105]. While well-studied in productive replication of numerous and diverse virus infections, surprisingly little is known regarding p53 function in the latent phase of herpesvirus infection. Certainly, prior studies suggest that some GHV latency proteins inhibit p53 (summarized below), but whether interactions between a GHV and p53 impact latency in a living host was previously unexplored. To the best of our knowledge, the data presented here represent the first in vivo evidence of an antagonistic relationship between host p53 and the GHV latency-associated transcription program that influences viral latency outcomes.

Since the absence of p53 correlated with increased GC proliferation and early viral latency establishment, our data strongly suggest that p53 is not simply inhibited by viral latency factors, an idea that has long been proposed in the GHV field, but that has become somewhat controversial in recent years. For instance, EBV nuclear antigen-1 (EBNA-1), a DNA-binding protein that facilitates maintenance of the latent viral episome, is expressed broadly in EBV cancers and hypothesized to facilitate cellular immortalization by promoting p53 proteolysis via an interaction with ubiquitin-specific protease 7 (USP7)[54,109]. However, despite the expression of EBNA-1 and in agreement with our findings, p53 is activated early during EBV infection of primary B cells, and LCLs remain responsive to p53 agonists[49,51]. Although the KSHV episome maintenance protein LANA inhibits p53 in overexpression systems[52,110,111], p53 remains functional in PEL cells that contain high levels of LANA protein[55,112]. While p53 can be inhibited by the MHV68 LANA homolog, the inhibitory function appears to be restricted to the lytic replication cycle, where it requires an undefined activating event and is assisted by additional lytic-phase viral proteins[57,104].

Given that inactivation of p53 by mutation or suppression is predisposing to cellular transformation, it is perhaps not surprising that viruses that establish lifelong chronic infections as part of their maintenance strategy would evolve to *not* block p53 function within latently infected cells. Presumably, potent inhibition of p53 would reduce host survival and limit viral transmission. However, it is important to also emphasize that this idea does not rule out the possibility that p53 responses are toned down by viral latency factors either transiently or subtly to enable GHVs to overcome p53-dependent barriers to latency establishment during initial colonization of host lymphoid tissues. It is additionally notable that p53 is frequently mutated in eBL[2], suggesting that p53 function is a critical determinant of pathogenesis for this disease. p53 knockout mice are primarily predisposed to thymic lymphoma and sarcomas[113], whereas people with p53 mutations exhibit a broad spectrum of cancers[114]. The differences in tumor spectrum between mice and humans are surely influenced by the types of p53 mutations present, but it is also possible that infectious agents influence the types of cancers that develop when p53 is absent or nonfunctional. Whether GHV infection influences the tumor phenotype of hosts when p53 is non-functional is unknown but could be tested in a p53-deficient setting using the mouse model, as we

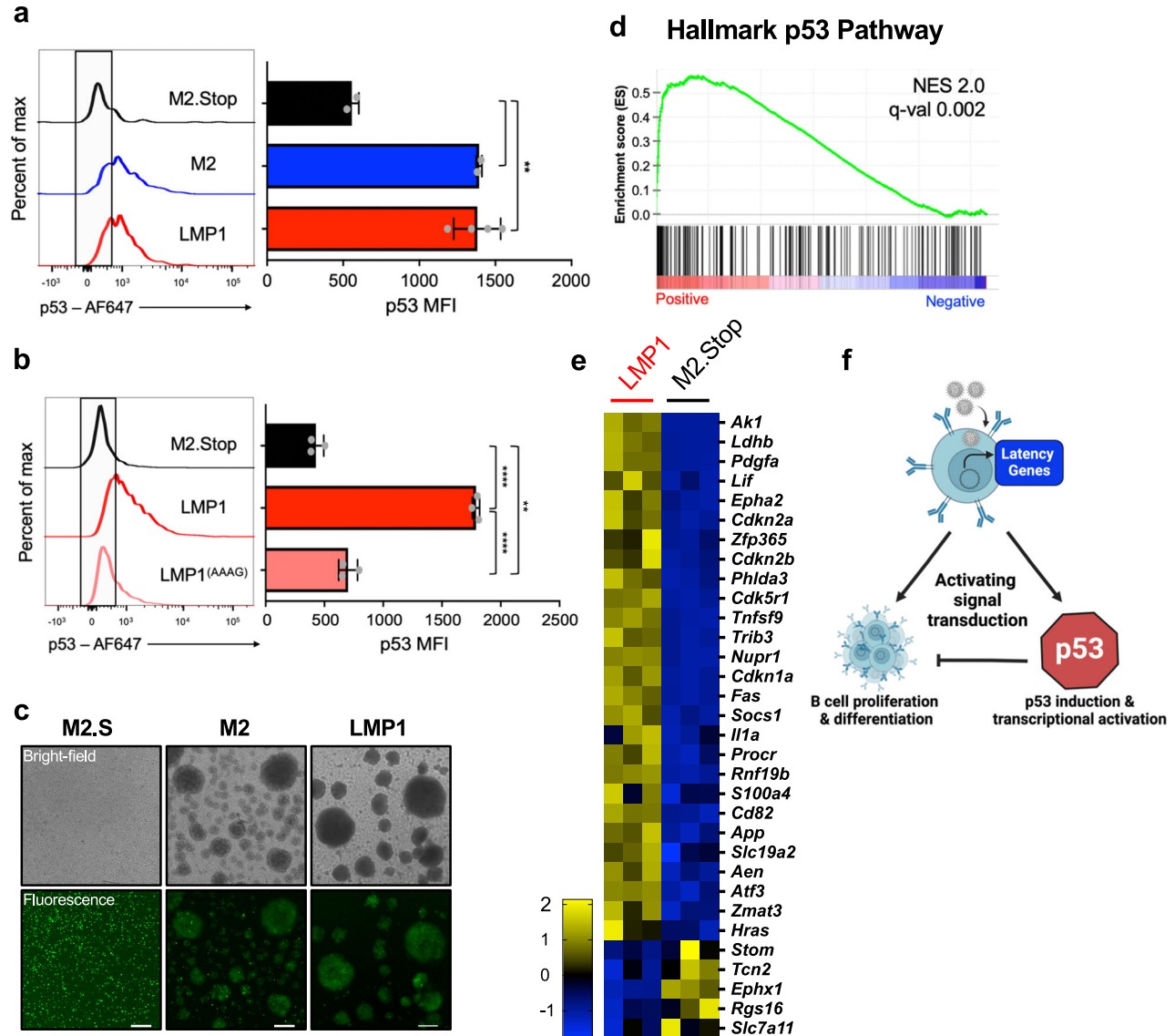

**Fig. 8 | EBV LMP1 activates p53 in mouse B cells. a, b** Flow cytometry analysis of p53 induction in primary murine B cells 4 days after transduction with LMP1, M2, LMP1 dominant-negative cell signaling mutant (LMP1^AAAG) or M2.Stop retroviruses. Representative histograms and mean fluorescence intensities of p53 staining in transduced cells are shown. Data represent means +/− SEM. Two-tailed Student's *t* test, **$p < 0.01$. The experiment was performed in technical triplicate. Gating strategies are shown in Supplementary Fig. 12. **c** Transduced p53^-/- B cells were imaged 48 h post-transduction. The scale bar indicates 250 μm. **d, e** RNA-seq was performed 4 days after transduction of primary B cells with LMP1 or M2.Stop (negative control) retroviruses to evaluate changes in transcription. Data show gene set enrichment analysis (GSEA) and hallmark gene z-score heatmaps comparing LMP1 to M2.Stop for the hallmark p53 gene set. The reference list was derived from Hallmark gene sets and compared with a pre-ranked list (by fold) of global average gene expression. Statistical scores are inset into the top right of analysis images. NES, normalized enrichment score. q-val, FDR-adjusted *p*-value. **e** z-score heatmaps show average expression data for LMP1 or M2.Stop expressing primary B cells. The genes presented were derived from GSEA and DEseq2 analysis of all genes with significant expression changes where the expression level increased from LMP1 to M2.Stop at least 1-fold and decrease from LMP1 to M2.Stop at least 1-fold. RNA-seq analysis of LMP1 transduced p53^-/- primary murine B cells and a table of hallmark gene sets with significant changes caused by LMP1 are shown in Supplementary Figs. 13 and 14. **f** Schematic of the model in which viral latency proteins that drive B cell proliferation and differentiation lead to the activation of p53. Lab, F. (2024) https://BioRender.com/b70y129.

suspect that enhanced B cell proliferation during MHV68 infection of p53-deficient animals is a hallmark of incipient lymphomagenesis.

With regard to a specific mechanism for p53 induction during GHV latency establishment, increased GC expansion during infection of p53 knockout mice supports a model in which viral latency proteins that drive B cell proliferation and differentiation lead to the activation of p53 (Fig. 8f). In agreement with known responses to viral oncogene expression[56,80,81], we suspected that p53 would then serve to counter-act the proliferative expansion of latently infected cells promoted by the viral latency program. Indeed, identification of the latency-specific gene product M2, which in and of itself promotes GC B cell

expansion[30,86], as a potent p53 activator in primary B cells directly supports this hypothesis. M2 serves as a molecular scaffolding protein that binds and activates Src-family kinases[27,28,87]. This leads to the induction of NFAT/calcineurin signaling pathways that promote IL-10 production and B cell differentiation[26]. Using M2 mutants and previously published inhibitors, we provide evidence that signal transduction via Src-family kinases is the key event for triggering p53 upon M2 expression, and not IL-10-related effects on the cell. It is well established that aberrant cellular activation and cell-cycle progression via oncogenes such as H-Ras and c-Myc triggers tumor suppressor responses, especially activation of p53-dependent inhibition of cell-

cycle progression[115,116]. Our data, therefore, support the conclusion that M2/Src activity is recognized by an infected B cell as a potential oncogenic threat, leading to p53 activity that restricts infected cell proliferation. Since p53 was equivalently induced in latently infected mice lacking a functional IFN-I receptor, we also conclude that p53 activation in this setting is not part of a traditional interferon-stimulated transcription response.

It will be of interest to determine if MHV68, like EBV[37,117,118], elicits ATM and/or ATM and Rad3-related (ATR) DNA damage responses as a mechanism of p53 phosphorylation and stabilization or if other kinases fulfill this role. For EBV, ATM and ATR are activated in response to a period of hyperproliferation and replication stress and limit the immortalization of infected B cells in culture[37,119,120]. The functions of ATM in MHV68 infection of mice have been evaluated, and MHV68's relationship with ATM is complex in that ATM expression in B cells does not restrict, but rather facilitates viral latency establishment[43,44]. A specific role for ATR, which is primarily activated by single-strand DNA breaks that occur as a result of DNA replication stress[121], would support the notion that aberrant cell-cycle progression and DNA replication caused by M2/Src is a stimulus for p53 activation. In further support of this notion, a previous RNA-seq analysis of MHV68-infected spleno-cytes revealed enhanced transcription of genes involved in the ATR signaling pathway, in addition to p53-related transcription[122]. While the ATR hypothesis is compelling, dissecting this pathway in vivo would be challenging, as ATR-deficient mice exhibit progeria syndrome and immune dysfunction[123].

Although M2 is unique to MHV68, it exhibits functional similarities to latency proteins encoded by human GHVs, such as EBV LMP1/LMP2A and KSHV K1/K15, especially integration of signaling events that occur upon B cell receptor and co-stimulatory molecule crosslinking[1,96,124–127]. Combined expression of EBV LMP1 and LMP2A in a transgenic mouse model caused GC B cell expansion and B cell differentiation into rapidly growing plasmablasts that have enriched expression of E2F targets and G2M checkpoint genes[128], which is analogous to M2 effects on B cells[86]. Both LMP1 and LMP2A mimic aspects of B cell signaling, but only LMP1 is a bona fide oncogene[36,40,99,129], which motivated our testing of LMP1-mediated p53 induction. Depending on the system being tested, LMP1 is reported to both induce p53-stabilization and cell-cycle inhibition or, conversely, to prevent p53 function[130,131]. Our data support the hypothesis that a p53 anti-proliferative response is activated in response to LMP1, where it prevents the outgrowth of transduced B cells. This interpretation is further supported by the observation that p53 stabilization is enhanced in lymphomas occurring in a transgenic mouse model in which LMP1 is constitutively expressed in B cells[99] and also by studies noting that LMP1 enhances p53 stabilization in cells expressing Large T antigen[100] and LMP1 expression levels strongly cor-relate with p53 levels in NPC[132]. In light of our results, we suspect that LMP1-driven B cell activation triggers p53 in an attempt to control tumor cell growth. While the percentage of cells with stabilized p53 was not evaluated in the transgenic mouse model[99], it is interesting to note that tumors were also characterized by NF-κB activation, which is necessary to overcome p53-mediated growth restriction during LCL formation[51]. Moving forward, it will be of interest to determine if LMP2A, K1, and K15 (and other human GHV latency proteins) also activate p53. Since LMP2A reduces LMP1 hyperactivation of B cells[133] and combined LMP1/LMP2A-deficient EBV exhibits diminished transforming capacity in a cord blood model of EBV lymphomagenesis[134], it will also be important to investigate whether LMP2A suppresses p53 induction by LMP1.

An important question that remains unresolved is how p53 can restrict early latency expansion, while also supporting long-term latency. This is further complicated by the counterintuitive observa-tion that pro-latency viral gene products serve as triggers for p53. In total, our data suggest that a dynamic exists between p53 and viral latency genes that is analogous to an automobile's gas pedal and brake – one accelerating the process while the other helps maintain

appropriate control (Fig. 8f). The cellular activating functions of the viral latency genes facilitate progression of the latently infected cell from point A (initial infection of a naïve B cell) to point B (progression through the GC reaction to enable long-term latency in memory B cells or reactivation from plasma cells). While beneficial to the virus in the long run, this is inherently a potentially dangerous method for pro-moting chronic infection as enhanced proliferative and/or differ-entiation signal transduction are recognized hallmarks of cellular transformation that should activate tumor suppressor responses in non-transformed cells.

In its most basic function, p53 is a cytoprotective molecule, pro-moting in response to a perceived threat the production of cell-cycle inhibitors that pause DNA replication and cellular proliferation until it is safe (genomically or metabolically) for such cellular processes to continue. If the threat persists or is catastrophic, p53 initiates an apoptotic cascade to eliminate the potential harm altogether[135]. Without the braking mechanisms provided by p53, we suspect that augmented cellular proliferation and differentiation driven by the viral latent transcription program results in enhanced DNA damage, accu-mulation of mutations, and, eventually, cellular transformation. Of note, we do find evidence of increased genomic instability in p53$^{-/-}$ mice after MHV68 infection (S.M. Owens and J.C. Forrest, manuscript in preparation), and the capacity to maintain in culture p53-deficient B cells expressing M2 and LMP1 (Supplementary Fig. 13b) suggests the potential for p53-dependent prevention of lymphomagenesis driven by certain GHV gene products.

While our work emphasizes the importance of p53 in limiting the GC expansion phase of MHV68 latency establishment, how the absence of p53 leads to a modest comparative reduction in latency over time, especially if p53 is activated by pro-latency viral genes, is not immediately apparent. It is generally thought that herpesviruses coevolved with their hosts yielding a virus-host relationship that allows the virus to chronically infect the host without causing major disease. This is perhaps reflected in our RNA-seq data, which indicate that p53 transcription is occurring simultaneously with M2-driven transcription related to B cell proliferation, survival, and differ-entiation. The early increase and late deficit in latently infected cell numbers suggest that p53 is a central player in this finely tuned balance that has evolved over millions of years. It is also interesting to note that M2, which is expressed in GC B cells, is an antigenic target for cytotoxic T lymphocytes (CTLs), as is LMP1[14,40]. While we did not detect any difference in adaptive immune responses with the reagents at our disposal, we have not ruled out the possibility that there is increased CTL clearance over time of M2-expressing B cells in p53$^{-/-}$ mice. As alluded to above, it also is possible that the absence of p53 cytoprotective functions results in an increased mutational load that eventually leads to cellular dysfunction and death. Another possibility is that p53 influences terminal B cell differentiation, and we acknowledge that we have not yet defined the cell types harbor-ing MHV68 during long-term infections of p53$^{-/-}$ mice. These ques-tions, as well as determining whether p53 restricts MHV68-driven B cell lymphomagenesis, highlight some of the important concepts and remaining questions resulting from our studies that could be tested in future experiments.

## Methods
### Cell culture and viruses
Swiss albino 3T12 fibroblasts were purchased from ATCC. Murine embryonic fibroblasts (MEFs) were harvested from C57BL/6 mice embryos and immortalized as previously described[72]. All cells were cultured in DMEM supplemented with 10% fetal bovine serum, 2 mM L-glutamine, and 100 μ/ml penicillin/streptomycin. Cells were main-tained at 37 °C in 5% $CO_2$. Viruses used in this study include WT MHV68[136], H2B-yellow fluorescent protein (YFP)-expressing MHV68[58], Cre-recombinase-expressing MHV68[64], LANA-beta-lactamase (73.Bla)-

expressing MHV68[59], and mLANA-null MHV68 (73.STOP)[16]. We generated an M2.Stop MHV68 on the 73.Bla backbone using *en passant* mutagenesis[137]. The MHV68 M2.Stop[93] targeting construct was generated by the insertion of a 26-bp linker into the SacII site within the M2 ORF at bp 4314 (oligonucleotides Oligo1 [5"AAGCTTAGGCTAGTTAA CTAGCCAGC] and Oligo2 [5"TGGCTAGTTAACTAGCCTAAGCTTGC]). The linker contained a diagnostic HindIII site. Oligo1 and Oligo2 were annealed and ligated into SacII-digested Lit38-M2. The addition of the oligomer resulted in a translational stop codon after residue 108 of the genomic M2 ORF. The γHV68M2.Stop construct was sequenced over the entire ORF with the Big Dye DNA sequencing kit (Applied Biosystems). A silent T-to-C mutation (that did not alter the predicted amino acid sequence but did result in the loss of a PstI site) at bp 4271 in the Lit38-M2.Stop construct was noted. Viral stocks were generated on 3T12 fibroblasts[57]. Viral titers were determined by MHV68 plaque assay[138].

## Mice and infections

All mice were housed and cared for according to the guidelines of the UAMS Department of Laboratory Animal Medicine and all state and federal requirements. Mice are housed in IVC cages in a room with a 12:12 light cycle and environmental conditions of 72°F and humidity of 30–70%. Both male and female mice are used for these experiments, and no overt differences in phenotypes due to sex are known to exist. p53-null mice on a C57BL/6 background (B6.129S2-*Trp53*[tm1Tyj]/J) were purchased from Jackson Laboratories and bred p53[+/-] x p53[+/-] to get a distribution of p53[+/+], p53[+/-], and p53[-/-] progeny. Ai14 (B6.Cg-Gt(ROSA) 26Sortm14(CAG-tdTomato)Hze/J) and AID-Cre (B6.129P2-Aicdatm1 (cre)Mnz/J) mice were purchased from Jackson laboratories and bred Ai14[+/+] x AID[Cre/Cre] to produce hemizygous offspring (Ai14[+/-]AID[Cre/+]). IFNAR1-null mice on a C57BL/6 background (B6/129S2-Ifnar1tm1Agt/ Mmjax) and Ai6 mice (B6.Cg-Gt(ROSA)26Sortm6(CAG-ZsGreen1)Hze/J) were purchased from Jackson Laboratories. 7–11-week-old mice were infected with 10⁴ PFU of WT MHV68, H2B-YFP MHV68, MHV68-Cre, WT or M2.Stop 73.Bla MHV68, or mLANA-null MHV68 by intranasal inoculation. Mice were sacrificed according to normal endpoint protocols. Blood was collected on days 0, 16, and 42 post-infection via the submandibular vein.

## Splenocyte isolation and limiting-dilution analyses

Limiting dilution analyses were performed with at least 2 independent infections with a minimum of 3 mice per group. Spleens were homogenized in a tenBroek tissue disrupter. Red blood cells were lysed by incubating tissue homogenate in 8.3 g/L ammonium chloride for 10 min at room temperature with shaking. Cells were filtered through a 40-micron mesh to reduce clumping. Frequencies of cells harboring MHV68 genomes were determined using a limiting-dilution, nested PCR analysis[72]. Frequencies of latently-infected cells capable of reactivating were determined using a limiting-dilution analysis for cytopathic effect induced on an indicator MEF monolayer[72].

## Antibodies, tetramers, and treatments

CD19-BV650 (6D5), IgM-BV421 (RMM-1), CD38-Pacific Blue (90), and purified B220 (RA3-6B2) were purchased from Biolegend (San Diego, CA). CD3e-PCPCy5.5 (145-2c11), CD8α-PCPCy5.5 (53-6.7), CD4-AF700 (RMA-5), B220-AF700 (RA3-6B2), GL7-eF660 (GL-7), CD38-PE-Cy7 (90), CD138-BV711 (281-2), NK1.1-PCPCy5.5 (PK136), and IgD-PE (11-26c) were purchased from eBioscience. Flow antibodies used at 1:300. Other antibodies include mouse anti-p53 Alexa Fluor 647-conjugate (Cell Signaling) used at 1:300, goat anti-GFP to enhance YFP detection (Rockland) used at 1:1000, anti-rat AF647-conjugate used at 1:400 and donkey anti-goat Alexa Fluor 488-conjugate (Jackson ImmunoResearch) used at 1:400. MHV68-specific MHC class I tetramers were generated by the NIH Tetramer Core. As a positive control for p53 induction, bulk splenocytes were treated with 10 grays of gamma radiation by exposure to a cesium-137 source. Following exposure, cells were allowed to recover at 37 °C in a tissue-culture incubator for 1 h prior to staining for flow cytometry.

## Flow cytometry

Cells were washed with FACS buffer (0.2% BSA, 1 mM in PBS) before blocking with Fc block (Invitrogen) and incubation with eF780 live/dead viability stain (eBioscience) for 10 min at 4 °C. Surface staining was then performed with antibodies diluted at 1:300 for 30 min incubation time at 4 °C. For intracellular stains, cells were fixed and permeabilized using a FoxP3 staining kit (eBioscience) following the manufacturer's guidelines. The data were collected using an LSRFortessa (Becton Dickinson) and analyzed using FlowJo (10.4.2) software.

## Immunohistology and microscopy

Tissues were fixed in 4% formaldehyde before paraffin embedding and cut into 5 μm sections. Tissue sections were deparaffinized, boiled in citrate buffer to retrieve antigens, blocked in BSA, and incubated with primary antibody overnight. Secondary antibodies were applied for 1 h, and slides were mounted using Vectashield[139]. All antibodies were utilized at 1:400. To neutralize high autofluorescence in spleen sections, sections were treated with TrueVIEW™ Autofluorescence Quenching Kit (Vector Laboratories) before mounting. Images were captured using a Keyence BZ-9000E.

## Real-time PCR and genotyping

Lungs were harvested from mice infected with H2B-YFP MHV68 on day 7 post-infection and homogenized using 0.5 mm silica beads in a Minibeadbeater (BioSpec)[72]. DNA was extracted from homogenized tissue using a DNeasy kit (Qiagen) according to the manufacturer's instructions followed by treatment with RNAse I (Qiagen) for 30 minutes at 37 °C. Quantitative PCR was performed on 500 ng of DNA with primers forward corresponding to the MHV68 ORF59 genomic locus (forward: 5'-ATG-CAG-ACC-TTC-CAG-CTT-GAC-3'; reverse: 5'-CTC-TTC-CAA-GGG-AGC-TTG-CG-3'). Cycling parameters were 95 °C for 30 s, 55 °C for 30 s, and then 72 °C for 30 s for 40 cycles on an Applied Biosystems StepOnePlus PCR system. Genotyping of mice was performed using a mouse genotyping kit (Kapa Biosystems). Briefly, DNA was extracted from mouse tail snips and amplified as previously described[113].

## p53 pathway transcriptional array

Ai6 mice were infected with 10⁴ PFU of MHV68-Cre intranasal inoculation and splenocytes were isolated at 16 days post-infection. One thousand ZsGreen positive and negative splenocytes were sorted in biological triplicate to a 96-well plate using a FACSAria II (BD). Transcriptome amplification was performed using the QIASeq Stranded RNA library kit (Qiagen). RT² Profiler PCR Array Mouse p53 Signaling Pathway (PAMM-027Z – Qiagen) was used for quantification. Reactions were performed in an Applied Biosystems StepOnePlus PCR system with cycling conditions of 10 min at 95 °C followed by 40 cycles of 15 s at 95 °C and 1 min at 60 °C. Biological triplets were analyzed using GeneGlobe RT² Profiler PCR Data Analysis software. Latent transcripts were analyzed by RT-PCR using inner primer sets as previously described[140]. Reactions were performed in an Applied Biosystems StepOnePlus PCR system with cycling conditions of 10 min at 95 °C followed by 40 cycles of 15 s at 95 °C and 1 min at 60 °C. Biological triplicate samples were analyzed in technical triplicate using the ΔΔCT method with β-actin as the cellular housekeeping transcript control. The results shown are technical triplicate for duplicate biological samples.

## Plasmids

Retrovirus plasmids were generated by restriction enzyme cloning of latency genes into pRetroX-IRES-ZsGreen (Takara). Latency locus

## Table 1 | pRetro_IRES_ZsGreen cloning primers

| | |
|---|---|
| M2-NotI-F | AGGTCAACTGCGGCCGCATGGCCCCAACACCCCCAC |
| M2-BamHI-R | ATTCCGGATCCTTACTCCTCGCCCCACTCC |
| M2-stop-NotI-F | AGGTCAACTGCGGCCGCATGGCCCCAACACCCCCACTCTAGAGATTCCCAATCCGTGGC |
| M11-NotI-F | AGGTCAACTGCGGCCGCATGAGTCATAAGAAAAGCGGG |
| M11-BamHI-R | ATTCCGGATCCTCAGACATAAATCACATTC |
| LANA-SacII-F | ATCAGCCGCGGATGCCCACATCCCCACCGAC |
| LANA-BamHI-R | ATTCCGGATCCTTATGTCTGAGACCCTTGTCCCTG |
| vCyc-XhoI-NotI-F | ATTCACTCGAGGCGGCCGCCATGGCCAGTCAAGAATTCCAAG |
| vCyc-BamHI-R | ATTCCGGATCCCTAGGCATTTATTTTGAAATAGTTTTCATC |
| LMP1_fwd | CGGATCTCACGTGGGCCCGCATGGAACACGACCTTGAG |
| LMP1_rev | GGATCCATCGATAGATCTGCTTAGTCATAGCCGCTTAG |

genes were amplified from WT MHV68-BAC DNA[136] and the frame-shift stop mutant, M2.Stop, was amplified from M2.Stop MHV68-BAC DNA[88] using primers in Table 1. LANA was cloned into the pRetroX vector via SacII and BamHI. vCyclin, M2, M11, and M2.Stop was amplified and cloned via NotI and BamHI. The LMP1 coding region was amplified from MSCV-N LMP1[141] (Addgene plasmid #37962) and cloned into pRetroX-IRES-ZsGreen via Gibson Assembly. M2-Tyr was amplified from a gene block (GeneWiz) and cloned into pRetroX-IRES-ZsGreen via NotI and BamHI. LMP1[AAAG] was generated as previously described. A variant 3′ primer was used to mutate codon 384 from Tyr to Gly. LMP1[G] was cloned into pRetroX-IRES-ZsGreen via Gibson Assembly. Site-directed mutagenesis was used to mutate the codons 204-208 from *PXGXT* to *AXAXA*. All plasmids were confirmed by sequencing (Plasmidsaurus).

### B cell cultures and retroviral transduction
Naïve B cells were isolated from spleens by immunomagnetic depletion using MojoSort Pan B cell Isolation Kit (Biolegend) and cultured in RPMI supplemented with L-glutamine, sodium pyruvate, penicillin-streptomycin, 50 µM 2-mercaptoethanol, and 10% fetal bovine serum. B cells were stimulated with 25 µg/ml LPS (Sigma). Retroviral supernatants were obtained from Phoenix cells (ATCC) 72 h after transient transfection (Lipofectamine LTX, ThermoFisher) with the retroviral plasmids and filtered through a 0.45 mm syringe filter. B cells were transduced 24 h after LPS stimulation with retroviral supernatants by centrifugation for 30 mins at $1000 \times g$ at 37 °C in the presence of 5 µg/ml polybrene. Cells were washed, and supernatants were replaced with RPMI. Triplicate wells per condition were analyzed by flow cytometry on day 4 post-transduction.

### Drug treatments
Cyclosporin A (500 µg/mL, TCI America), PP2 (10 µM, Selleck Chemical LLC), and PP3 (10 µM, Tocris Bioscience) were reconstituted in DMSO. For antibody treatments, mouse IgG Isotype Control (20 µg/mL, ThermoFisher) and anti-mouse IL-10R (20 µg/mL, BioXCell) were diluted in cRPMI. The drugs were used at a final concentration as indicated in parentheses above.

### EdU incorporation assays
Mice were injected with 100 µg of 5-ethynyl-2′deoxyuridine (EdU, Invitrogen) 2 h prior to splenocyte harvest. Cells were fixed with 4% paraformaldehyde in PBS and Click-iT chemistry (BD Pharmagen) was performed according to the manufacturer's instructions to fluorescently mark EdU + cells.

### Caspase 3/7 cleavage assays
Apoptotic cells were labeled using fluorescently labeled inhibitors of caspases (FLICA 660, Immunochemistry Technologies, LLC). Briefly, splenocytes were stained with FLICA 660 for 1 h at 37 °C. Following

incubation, cells were washed and surface stained for GC B cells markers as described above.

### Enzyme-linked immunosorbent assays
Viral antigen was prepared by infecting 3T12 fibroblasts at an MOI of 0.5 PFU/cell for 96 h. Cells were washed and fixed in 1% PFA. Viral antigen was used to coat high-binding plates (Denville) overnight at 4 °C. After blocking in 5% FBS in PBS, serially diluted serum was incubated on the plate, followed by incubation with HRP-conjugated IgM or IgG Ab (Southern Biotech). SureBlue substrate (KPL) was added to detect the Ag-specific Abs. The wells were read on a FLUOstar Omega plate reader (BMG Labtech).

### Ultra-low input RNAseq
Following 73.Bla MHV68 infection, MHV68 + cells are defined by cleavage of CCF4-AM fluorescent substrate[59]. Briefly, draining lymph nodes from infected animals were collected and homogenized as described above. β-lactamase activity was detected using the Live-BLAzer FRET-BG/Loading Kit with CCF4-AM (Thermo Fischer Scientific #K1095). Cells were incubated for 1 h at room temperature, protected from light, washed 3x with FACS buffer, and immediately subjected to fluorescent-activated cell sorting (FACS) performed on a FACS Aria flow cytometer (BD Biosciences). Sorted cells were frozen in Trizol. The library was constructed by GeneWiz (Azenta) using Illumina RNA prep with enrichment for full-length transcripts. Polyadenylated transcripts were reverse transcribed using SMART-Seq® v4 Ultra® Low Input RNA Kit (Clontech), followed by second-strand synthesis of full-length cDNA and amplification of the product. Illumina®-compatible sequencing libraries were constructed and then validated for sufficient concentration and appropriate fragment sizes. Samples were sequenced on the Illumina HiSeq® 2500 with a 1 × 100 bp single-end configuration. Data processing and analysis are described in RNAseq methods.

### RNAseq
Transduced B cells were collected 4 days post-transduction. RNA was harvested using Trizol. The library was constructed by BGI Laboratories using DNBSEQ Eukaryotic transcriptome protocol. Following mRNA isolation, the mRNA was fragmented, and first-strand synthesis performed. Double-stranded cDNA fragments were subjected to end-repair, and a single 'A' nucleotide was added to the 3′ end of the blunt fragment. Following library QC, single-strand cDNAs were circularized and replicated via rolling cycle amplification. DNA nanoballs were generated and loaded into patterned nanoarrays using high-intensity DNA nanochip technique and sequenced through combinatorial Probe-Anchor Synthesis (cPAS). Sequencing was performed on a DNBSEQ-G400 (BGI Genomics Co). Raw sequence reads were quality-checked using fastQC and trimmed based on quality (Phred > 30). After trimming, reads < 30 bp in size were discarded. Trimmed

sequence reads were mapped to reference genome mm10 using STAR aligner with default parameters[142]. The gene count tables were extracted from alignment results using bedtool2 software[143]. Read counts were normalized using the Voom method[144]. Normalized read counts were analyzed using DEseq2[145]. Representative gene set enrichment analysis (GSEA)[146,147] was performed using the reference list derived from Hallmark gene sets[148] and compared with a pre-ranked list (by fold) of global average gene expression.

## Statistical analysis

All data was analyzed using GraphPad Prism software (GraphPad Software, http://www.graphpad.com, La Jolla, CA). MFI was analyzed by one-way ANOVA. The total number of immune cells subset per animal were assessed using a Mann-Whitney unpaired two-sided T-test. Based on the Poisson distribution, the frequencies of the viral genome–positive cells and reactivation were obtained from the non-linear regression fit of the data where the regression line intersected 63.2%. Extrapolations were used for samples that did not intersect 63.2%. Non-linear regressions were assessed by the Extra sum-of-squares F test of the LogEC50. PCR and reactivation assays were performed with at least 2 independent infections with a minimum of 3 mice per group. Edu$^+$ and active caspase cell frequencies were analyzed using two-tailed Student's $t$ tests. All results are insignificant unless otherwise noted. Dots represent individual animals.

## Reporting summary

Further information on research design is available in the Nature Portfolio Reporting Summary linked to this article.

## Data availability

Ex vivo B cell transduction mRNAseq data generated in this study are in the NCBI database under GEO Accession # GSE225579. (https://www.ncbi.nlm.nih.gov/geo/query/acc.cgi?acc=GSE225579). In vivo lymphocyte mRNAseq data generated in this study are in the NCBI database under GEO Accession # GSE285085 Source data are provided in this paper.

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

## Acknowledgements

We thank Andrea Harris for the UAMS Flow Cytometry Core. This work was supported in part by grant R01CA167065 from the NIH National Cancer Institute and start-up funds from the UAMS College of Medicine and Arkansas Biosciences Institute to J.C.F. S.J.M. was supported through a diversity supplement to R01CA167065 and R25GM083247 from the National Institute of General Medical Sciences (NIGMS) of the National Institutes of Health. The Flow Cytometry and Microscopy Core and the work described here were also supported in part by the Center for Microbial Pathogenesis and Host Inflammatory Responses award P20GM103625 from the NIH National Institute of General Medical Sciences Centers of Biomedical Research Excellence. S.M.O. was supported by Translational Research Institute (TRI) grant TL1TR003109 through the NIH National Center for Advancing Translational Sciences. M.M. is supported by K22CA241355 from the National Cancer Institute with additional support from P30GM145393 from the National Institute of General Medical Sciences. The funders had no role in study design, data collection, and interpretation, or the decision to submit the work for publication. The model in Fig. 8 was created in BioRender. Lab, F. (2024) https://BioRender.com/b70y129.

## Author contributions

S.M.O., J.M.S., and J.C.F designed the study, analyzed data, and wrote the manuscript. G.L. constructed and tested retroviral vectors. S.J.M. performed *IFNAR1* experiments, analyzed data, and wrote the manuscript. E.S. participated in the study design. D.G.O. generated MHV68 BAC recombinants. I.N. performed RNA-seq raw data analysis. M.M. performed RNA-seq pathway analyses and wrote the manuscript. D.G. and J.S. assisted with flow cytometry data analysis. J.C.F. acquired funding. All authors read and approved the final manuscript.

## Competing interests

The authors declare no competing interests.
