## [Transparent Peer Review file · Nature Communications]

Intrinsic p53 Activation Restricts Gammaherpesvirus-Driven Germinal Center B Cell Expansion during Latency Establishment

Corresponding Author: Dr James Forrest

Version 0:

Reviewer comments:

Reviewer #1

(Remarks to the Author)

This is an important study that defines the role of p53 in the context of chronic gammaherpesvirus infection. The use of the MHV68 experimental system uniquely allows the authors to address this knowledge gap. They identify latent protein M2 and a functional orthologue LMP1 as inducers of p53 activity, at least in the context of primary B cell cultures. While this is a novel observation, a major knowledge gap remains in this study. Specifically, if M2 indeed increases p53 expression and activity, which appears to be detrimental for the latency establishment, why is this evolved feature happening, including for LMP1? How is the seemingly antiviral function of M2 in p53 induction cross-play with the major role of M2 in supporting establishment of MHV68 latency in the context of chronic viral infection in vivo and B cell proliferation upon lentivirus transduction in culture? The authors do not close the logic loop, as there are no studies defining M2-p53 interaction in the context of infection or the M2.stop virus mutant biology in p53 KO mice.

Additional major concerns:

Fig. 2, 3 Data presented here are either generated using total splenocytes or infection is quantified in B cells, along with the overall characterization of B cell response. While this is a reasonable approach, the authors do not demonstrate any data regarding the latent infection of non-B cells. This is particularly relevant for p53KO mice, given that by 9 weeks of age there are detectable T cell clones with an abundance several orders of magnitude higher than any T cell clone in age matched control mice. Even B cell-tropic gammaherpesviruses can infect T cells and it is possible that some of the T cell clones that are arising in p53 mice support the infection and alter its parameters, particularly during long-term infection when the aged p53 KO mice have progressive thymomas.

Fig. 2. It is remarkable that a 14-fold increase in the frequency of genome positive cells was accompanied by minimal if any increase in reactivation. Given that plasma cells almost exclusively reactivate MHV68 in the spleen, are the infected B cells not able to differentiate into the plasma cells in the absence of p53? Or are plasma cells incapable of supporting efficient MHV68 reactivation without p53?

There are discrepant findings in the gene expression analyses from infected splenocytes in Fig. 1 and M2-transduced B cells in Fig. 5/6, with the comparison itself being apples to oranges, as M2 is thought to be primarily expressed in B cells. Which ones of the differentially expressed genes in Fig. 1 are actually downstream of p53? How do the levels of lenti-driven M2 expression compare to the ones in infected cells? Expression of the same p53 target genes should be compared in YFP positive vs. negative B cells from infected animals and presented alongside with the differential expression of these genes in control vs. M2-transduced B cells, with addition of p53 KO control.

Data in Fig. 7 are exciting but also caveated. The authors put forth a significant effort to highlight the biological similarity between the two viral genes that do not share any amino acid homology, providing evidence for an important concept that diverse viral proteins can have overlapping biological functions. However, LMP1 may have species-specific protein interactions in B cells that would not be recapitulated in murine B cells. Given the well-developed lenti-based transduction of LMP1, the key observation of p53 increased protein levels and activity should be repeated in human primary B cells, in addition to transduction of other EBV genes that are biological mimics of negative controls the authors used in Fig. 5a.

Minor:

For all in vivo studies, including in supplemental data, the authors should state the number of animals/independent experiments that the data were pooled from.

Fig. 2b: please adjust the Y-axis label for clarity (percent vs. absolute number)

Fig. 2e: the quality of images, particularly for B220 and low power YFP staining is poor. It is not clear why the background color for the same stain is different (blue vs. green for B220, etc).

Long-term latency data should be incorporated into the main manuscript.

Fig. 4B. Annexin V is not a reliable marker of apoptosis for primary cell types. Apoptosis should be probed with reagents that directly measure caspase 3 activation.

Supplemental figure 4 legend is mislabeled

MHV68-Cre has a tainted history in the field, as it is variably attenuated in wild type mice. Although supplemental Figure 1 does LD-PCR-based validation of the latent reservoir for this mutant, it is unclear how many mice/independent biological replicates were analyzed. Given the use of this suspect viral mutant in Fig. 1D, E, the observed changes need to be validated with the bla reporter MHV68 made by Tibbetts group if the authors would like to pursue two independent marking strategies.

Reviewer #2

(Remarks to the Author)

The manuscript "Intrinsic p53 activation restricts gammaherpesvirus-driven germinal center B cell expansion during latent establishment" by Owens et al., reported that: p53 is activated in B cells infected with MHV68; that p53 limits MHV68 latency; and that the latent M2 protein activates a p53-dependent anti-proliferative response through a Src-mediated mechanism. Finally, they demonstrate that the EBV LMP1 protein also induces a p53 response in primary B cells. Overall, some findings are interesting; however, there are important concerns and questions that need to be solved. The lack of information makes difficult to interpret some of the results.

The authors propose that the "p53 stabilization" is not a consequence of viral infection, it represents a cell-intrinsic reaction to the infection with MHV68.

What are the mechanisms that prevent the activation of p53 as a general consequence of viral infection?

Why do the authors say "p53 stabilization"? They do not evaluate p53 stability. Acute viral replication was evaluated in lung whereas latent one was analyzed in spleen. Do the results depend on the tissue analyzed? To rule out a tissue effect, the authors should evaluate latency and acute replication in other tissues. Does acute replication of MHV68 not induce p53? Do the authors detect interferon production by the infected cells? If this is the case, an increase in the levels of p53 protein by the uninfected cells is expected.

Why the authors do not compare the same amount of latently infected splenocytes derived from p53 ko and WT mice for being reactivated?

It has previously reported that MHV68 infection triggers a DNA damage-mediated induction of p53 but when the viral cycle progresses several viral proteins inhibit the p53 pathway. How are these results explained in your model?

It has been reported that p53 enhances IFN signaling, specifically the induction of genes containing IFN-stimulated response elements and IFN gamma secretion. Do the levels of interferon produced in response to MHV68 infection differ between p53 WT and p53 ko mice? could this have an impact on the number of cells infected in the mouse?

Many graphs do not show statistical significance.

Fig 1. Some graphs do not have standard deviation and it is not clear how many samples were evaluated. In addition there is no information about the statistical analysis of some graphs.

Fig.2C. There are significant differences between p53 WT and KO cells; however, it is unclear whether this lack of significance is due to the high variability. A more detailed information of the methodology and results about how the authors did the score of cytoplasmic effect would be helpful.

Fig.2D. The immunohistochemical images are difficult to compare due mainly to the different B220 signal in p53WT and KO mice. The authors mention that the B220 staining was much higher in the spleens of p53 KO mice than in wt mice. Why is this happening? It has been previously reported that M2 expression leads to a downmodulation of B220. How are these results explained in your model?

Fig.3A. Does MHV68 infection significantly increase the total number of splenocytes exclusively in p53 KO mice?

Fig. 4a. Is there only two mock samples? More samples are needed to interpret the results, specially when there is so high deviation in the data.

Fig 4b. How many samples were evaluated? The authors did not observe statistical differences between p53WT and p53KO; however, in this case the deviation is reduced and the difference is around 10%, therefore it is hard to believe there is no significant differences.

Fig. 5. Why experiment a was done at 4 days after transduction and B at 48 h after transduction?

Fig. 6a. Is M2.Stop expression leading to an increase in the percentage of Edu+ in p53 KO cells?

Figure 7 b. The authors show GFP expression at 48 h after transduction with M2.stop, M2 or LMP1: the cells look totally different and the authors state that M2 or LMP1 expression leads to a large cluster of cells. It would be helpful to show the level of expression of the different proteins at to know at what point after transduction the clusters begin to form.

Analysis of the viral latency and viral proliferation in p53KO and p53 wt cells treated or not with Src inhibitors would help to evaluate the consistence of their hypothesis.

Reviewer #3

(Remarks to the Author)

In this study, Owens and Sifford, et al. employ a suite of techniques coupled with a mouse model to demonstrate in vivo that p53 is activated during latency establishment of the gammaherpesvirus MHV68. By leveraging a targeted sufficiency screen, they identify that the herpesvirus protein M2 is sufficient to stabilize p53 and to cause an anti-proliferative response. Using small molecule approaches, they uncover that this response is mediated by M2-induced activation of a Src-family kinase. The authors demonstrate functional conservation among gammaherpesviruses by expressing the EBV protein LMP1, the ortholog of M2, in mice and showing that it is sufficient to drive the same effects on p53. Overall, the paper is logical in its presentation and flow and has a number of noteworthy findings. However, I have 3 major along with a number of minor concerns outlined below. Addressing these points would strengthen the manuscript.

Major Points:

1. The authors nicely demonstrate in Figure 5a that M2 expression leads to increased p53 abundance. However, they do not quantify the levels of M2 expression compared to the other MHV68 proteins that they express in these tissues. Western blot or RT-qPCR analysis should be done to demonstrate similar levels of expression of all of the viral proteins that they are introducing, to confirm that the changes in p53 that they see are due to M2 and not a consequence of M2 being the only well-expressed protein. Also, in 5b, is there a significant difference between PP3 and PP2 treated tp53 ^{-/-} cells? Please indicate whether there is or is not, as in the PP3 and PP2 treated tp53 ^{+/+} cells on the left.
2. The authors demonstrate that M2 promotes p53 stabilization via a Src-family kinase, though uses a large dose of the PP2 inhibitor (10 mM), which raises the possibility that there could be off-target effects. They do try to control for this with the PP3 drug, but it may have a distinct off target profile. The paper would be enhanced if the authors could show that knockdown of a specific SRC kinase produces a similar phenotype, but I realize this may be difficult to do together with M2 expression in primary cells in the timescale of the experiment. However, could it be done at a later timepoint? For instance, first express M2, then perform the shRNA control or src kinase knockdown on the proliferating tp53 ^{+/+} vs ^{-/-} cells and analyze EdU several days later? Alternatively, can the authors demonstrate similar results with a small molecule antagonist specific for a particular SRC kinase, such as Lyn, or at least confirm with SRC inhibitors that are active at much lower doses? Or use an M2 or even LMP1 mutant that is deficient for SRC kinase activation?
3. The authors cover several pathways and proteins in this study. Including a final model that summarizes their key findings would be helpful for the reader to synthesize everything what they have presented.

Minor Points:

1. In Figure 1a, the authors label that only 1.02% of the cells are infected and YFP+. However, this number seems low based on the rest of the data. Can the authors please explain this discrepancy?
2. Figure 1c highlights the increases in gene expression for p53-responsive genes. While this is evident for p53 target Cdkn1a, there is a very modest (?1 fold) change in Mdm2. This does not seem to support the argument that p53 signaling is active. Would qPCR for additional p53 target(s).
3. In line 161, define what tp53 is. Is this murine tp53? Also, In Figure 1e, can the authors please note how many replicates are represented by the data?
4. In Figure S3, data is shown for both 16 and 42 days post-infection. However, there is no description in the text as to the significance of these two time points. Please add some text to indicate the importance of these two time points.
5. In Figure S4, panel a should be panel d based on the figure legend, and panels b,c,d should be a,b,c. Please adjust the figure or the figure legend so that they match.
4. For clarity, it would help if the reactivation shown in Figure S5 could be shown as a percentage as it is shown in Figure 2. Also, Figure S5 shows that day 42 viral burden is lower in tp53^{-/-} mice, despite the finding that there is increased proliferation of infected cells at day 16. How can the authors reconcile this discrepancy? Is there an enhanced immune response that restricts viral infected cells between days 16 and 42, which could be an adaptive response that took time to develop? Or is there enhanced death or perhaps differentiation of infected cells after day 16?
6. Can the authors please explain how they selected the log2 fold change cutoffs for their GSEA analysis, especially since the increase fold change cutoff is 2 but the decrease cutoff is only 1.5?
7. In Figure 5a, the frameshift stop mutant is not shown. However, it is discussed in the text. Can the authors please remedy this discrepancy?
8. The legend for Figure 6 refers to panels c-e, but there are only panels c-d.

9. In Supplementary Figure 11, why do the MYC targets go up in p53^{-/-} M2 but have the opposite trend in the LMP1 dataset?
10. It looks like the gating strategy for Figure 7 is shown in Supplementary Figure 10, not 11.
11. It would help with the clarity if the authors could refer to specific supplementary figure panels instead of the entire supplementary figure when possible.
12. In line 484, "grays" should probably be "rays." Please check this.
13. PMID 18242176 reported that LMP1 upregulates p53, would cite and discuss. Likewise, PMID 15330177 correlated LMP1 and p53 expression in NPC.
14. Line 372, would consider modifying the text to indicate that people with tp53 mutations (Li Fraumeni syndrome) have just one copy of p53 mutated, as opposed to tp53 KO mice.

Version 1:

Reviewer comments:

Reviewer #1

(Remarks to the Author)

The revised manuscript is significantly improved, with only one comment remaining from this reviewer:

Figure 2. While the authors make a statement in the rebuttals that viral reactivation is increased in p53 deficient mice, the data in Fig. 2c do not seem to support this statement, as the SEM error bars are largely overlapping and statistically significant difference is not indicated. Similarly, data in Fig. 2b should be analyzed statistically, with additional cell dilutions analyzed by LDPCR for the p53 deficient splenocytes, as the intersection with the dotted line for this data set is derived from extrapolated curve. The authors present convincing data using marked reporter viruses; however, limiting dilution-based assays are the gold standard of the field and should complement the conclusions reached using reporter MHV68.

Reviewer #2

(Remarks to the Author)

This study analyzes the role of p53 in latent gammaherpesvirus infection using MHV68 as an experimental model. The authors reveal that the latent protein M2 of MHV68 induces p53 activity and this activation limits B cell proliferation and differentiation and the latency establishment. They demonstrate functional conservation by revealing that the EBV LPM1 protein exerts the same effects on p53. The work is original and is of significance to the field. The work supports the main conclusions of the article. The methodology is appropriate to the research question.

Reviewer #3

(Remarks to the Author)

The authors have responded to each of my concerns. I feel that the manuscript has improved and is now acceptable for publication in Nature Communications.

Version 2:

Reviewer comments:

Reviewer #1

(Remarks to the Author)

I appreciate the author's thorough response and the technical complexity of the limiting dilution assays in the MHV68 field. Given the high SEM associated with the pooled data, data from individual experiments (limiting dilution assays pooled in the main figure) should be presented in supplemental figure.

Other than inclusion of data from individual LD-based experiments in supplemental figures, I don't have any concerns.

RESPONSE TO REVIEWER COMMENTS:

We would like to begin by thanking the reviewers for their time, opinions, and insightful comments and suggestions. To address concerns raised by the reviewers about the original submission, we have performed several new experiments and modified the text of the manuscript. Major new experiments include:

- 1. Analyses of p53 induction during MHV68 latency establishment in type I interferon receptor knockout mice.*
- 2. RNA-seq comparing transcription profiles of germinal center B cells sorted from mice infected with either M2-null or WT MHV68.*
- 3. Evaluations of p53 induction in primary murine B cells expressing mutants of M2 and LMP1 that do not activate signaling cascades.*

We have also included additional controls and updated analyses of previous experiments to address other questions, comments, and concerns raised by the reviewers. Importantly, we have extensively modified the discussion to clarify our interpretation on how the interplay between M2 and p53 might function in chronic MHV68 infection. We have addressed each and every comment, and, in our opinion, the new experiments, additional controls, and text modifications have strengthened our original interpretations and conclusions, making for a much more thorough and complete evaluation of p53 function in gammaherpesvirus latency establishment. Responses to specific reviewer comments follow below.

Reviewer #1 (Herpes Latency) (Remarks to the Author):

This is an important study that defines the role of p53 in the context of chronic gammaherpesvirus infection. The use of the MHV68 experimental system uniquely allows the authors to address this knowledge gap. They identify latent protein M2 and a functional orthologue LMP1 as inducers of p53 activity, at least in the context of primary B cell cultures. While this is a novel observation, a major knowledge gap remains in this study. Specifically, if M2 indeed increases p53 expression and activity, which appears to be detrimental for the latency establishment, why is this evolved feature happening, including for LMP1? How is the seemingly antiviral function of M2 in p53 induction cross-play with the major role of M2 in supporting establishment of MHV68 latency in the context of chronic viral infection in vivo and B cell proliferation upon lentivirus transduction in culture? The authors do not close the logic loop, as there are no studies defining M2-p53 interaction in the context of infection or the M2.stop virus mutant biology in p53 KO mice.

Response: Thank you for your comments and suggestions. We agree that this is an important point and apologize for not adequately addressing this seemingly counterintuitive observation in the previous submission. We have extensively revised the discussion in an attempt to clarify our interpretations of how pro-latency viral factors support latency establishment while also activating p53. In short, we consider this a classic oncogene and tumor suppressor interaction in which viral proteins that promote nonphysiologic cellular activation and differentiation trigger p53, which primarily functions as a cytoprotective molecule that likely limits potential damage to the cell in a manner that benefits the virus in the long-run. We emphasize that herpesviruses have coevolved with their hosts to a point where chronic infection is a finely tuned balance between viral gene product functions and host responses. Defining the mechanism(s) of reduced latency over-time in p53-deficient mice is a focus of ongoing experimentation in our laboratory, but we consider these studies the foundation of a stand-alone follow-up story, as the current manuscript focuses primarily on p53-mediated restriction of latency establishment and not long-term maintenance of latent infection. To provide additional in vivo relevance to the

current study, we performed new experiments (as suggested below) comparing transcription in WT and M2-null MHV68 infected GC B cells, finding that p53-related transcription does indeed occur in a M2-dependent manner in vivo (new Fig. 6). This occurs at the same time that the pro-latency transcription program involving IL-10 and IRF4 is upregulated by M2. We elaborate on these findings, our interpretations, and describe testable downstream hypotheses in the discussion.

Additional major concerns: Fig. 2, 3 Data presented here are either generated using total splenocytes or infection is quantified in B cells, along with the overall characterization of B cell response. While this is a reasonable approach, the authors do not demonstrate any data regarding the latent infection of non-B cells. This is particularly relevant for p53KO mice, given that by 9 weeks of age there are detectable T cell clones with an abundance several orders of magnitude higher than any T cell clone in age matched control mice. Even B cell-tropic gammaherpesviruses can infect T cells and it is possible that some of the T cell clones that are arising in p53 mice support the infection and alter its parameters, particularly during long-term infection when the aged p53 KO mice have progressive thymomas.

Response: This is an interesting point. We reevaluated infection of T cells and found no YFP+ CD8 T cells and only minimal YFP+ CD4 T cells that did not differ between WT or p53 KO mice. These data are now included in Supplementary Fig. 6. With regard to expansion of T cell clones, it is important to note that we did not detect an increase in T cell numbers in our experiments, and virus-specific T cell responses were identical in both WT and p53 KO mice (Supplementary Fig. 5).

Fig. 2. It is remarkable that a 14-fold increase in the frequency of genome positive cells was accompanied by minimal if any increase in reactivation. Given that plasma cells almost exclusively reactivate MHV68 in the spleen, are the infected B cells not able to differentiate into the plasma cells in the absence of p53? Or are plasma cells incapable of supporting efficient MHV68 reactivation without p53?

Response: Thank you for this comment; we apologize if our description of these data was unclear. We do detect increased reactivation (Fig. 2c,d – WT 1/11,500 vs. p53^{-/-} 1/2,500), however it correlates almost exactly to the increase in the frequency of infected splenocytes (See Fig. 2a,b,d). Therefore, there is not an increase in the reactivation efficiency, per se. Additionally, while the increase in the number of plasmablasts in p53 KO mice was not statistically significant, there were more CD138⁺B220^o cells present in p53 KO spleens, and the frequency of these cells that were infected (YFP+) was significantly greater, which also agrees with the increase in the frequency of reactivating cells in p53^{-/-} mice.

There are discrepant findings in the gene expression analyses from infected splenocytes in Fig. 1 and M2-transduced B cells in Fig. 5/6, with the comparison itself being apples to oranges, as M2 is thought to be primarily expressed in B cells. Which ones of the differentially expressed genes in Fig. 1 are actually downstream of p53? How do the levels of lenti-driven M2 expression compare to the ones in infected cells? Expression of the same p53 target genes should be compared in YFP positive vs. negative B cells from infected animals and presented alongside with the differential expression of these genes in control vs. M2-transduced B cells, with addition of p53 KO control.

Response: Thank you for this comment. As the reviewer correctly asserts, the analyses performed in Figures 1, 5, and 6 were indeed different. Transcription data in Figure 1 represent a targeted RT-PCR array comparing sorted MHV68 infected cells (which to clarify were ~96% CD19⁺B220⁺, see Supplementary Fig. 2) to uninfected CD19⁺B220⁺ B cells. As highlighted in Fig. 1c, the infected cells express the full complement of latency-associated transcripts, not just M2, and exhibit increased expression of many (not all) targets present in the array. This was

*designed as an initial experiment to determine if p53-responsive transcription was upregulated in latently infected cells, since we thought it possible that p53 was induced, but functionally inhibited by latency gene products, which would have agreed with the GHV dogma that latency genes inhibit p53. We viewed the increase in p53-related transcription as rationale for pursuing infections of p53 KO mice, and we consider this an important contributor to the logic flow of the manuscript. These data are also consistent with previous RNA-seq evaluations from the Virgin lab that noted enhanced transcription of p53-related transcripts in spleens of mice after MHV68 infection, also including *cdkn1a* and *mdm2* (Canny et al. JVI 2014). In light of the profound phenotypes observed for MHV68 latency establishment in p53-deficient mice, we emphasize that the retroviral transduction experiments described in Fig. 5 and current Fig. 7 (previously Fig. 6) were an attempt to define which (if any) specific latent viral gene products were in-and-of-themselves sufficient to activate p53. RNA-seq was performed as an unbiased evaluation to identify M2- (and LMP1) dependent transcriptional pathways, and we performed the analyses in p53-deficient cells at the same time to define the p53-dependent transcriptome. All of that said, we do agree that the direct comparison back to infected cells from mice felt like a missing piece in our interpretations and conclusions. We therefore performed new in vivo RNA-seq experiments in which we isolated cells infected with either WT or M2-null MHV68 (using the Beta-lactamase reporter MHV68 as suggested below) and find that the presence of M2 does indeed correlate with induction of p53 hallmark genes in infected mice (new Fig. 6). We have highlighted in red the congruent p53-related transcripts present in both MHV68 infected cells in mice and M2 transduced primary B cells in culture.*

Data in Fig. 7 are exciting but also caveated. The authors put forth a significant effort to highlight the biological similarity between the two viral genes that do not share any amino acid homology, providing evidence for an important concept that diverse viral proteins can have overlapping biological functions. However, LMP1 may have species-specific protein interactions in B cells that would not be recapitulated in murine B cells. Given the well-developed lenti-based transduction of LMP1, the key observation of p53 increased protein levels and activity should be repeated in human primary B cells, in addition to transduction of other EBV genes that are biological mimics of negative controls the authors used in Fig. 5a.

Response: We agree, and we noted this potential caveat in the discussion section of the manuscript, while also highlighting the previous publications that demonstrate p53 induction in LMP1 transgenic mice, during EBV infection of primary human B cells, and the capacity of p53 to limit LCL formation. We assert that the intention of the LMP1 experiments we present was to highlight a parallel between functionally similar GHV latency proteins, while offering a potential explanation for p53 induction during EBV infection of B cells that is supported by previous publications. With those points in mind, it is our opinion that defining the capacity of LMP1 to induce p53 in retrovirally transduced human B cells is an important experiment in a larger, stand-alone study that encompasses WT and LMP1 mutant EBV infections and molecular pathway mapping – experiments that we plan to pursue in future studies.

Minor:

For all in vivo studies, including in supplemental data, the authors should state the number of animals/independent experiments that the data were pooled from.

Response: The figure legends have been updated to note that each graphed dot represents single data points from either pooled samples or individual mice or infections. Further details are provided in the Methods section.

Fig. 2b: please adjust the Y-axis label for clarity (percent vs. absolute number)

Response: The Y-axis label has been updated to include percent.

Fig. 2e: the quality of images, particularly for B220 and low power YFP staining is poor. It is not clear why the background color for the same stain is different (blue vs. green for B220, etc).

Response: We agree, and we have repeated these experiments using immunofluorescence microscopy. Staining of mock-infected spleens and additional images from infected mice are included in new Supplementary Fig. 4.

Long-term latency data should be incorporated into the main manuscript.

Response: We agree on the importance of the long-term latency data, but would like to keep the main body and primary figures of the manuscript focused on the early latency response at day 16 post-infection. In general, the reduction represents a rather modest reduction in cells harboring viral genomes for these time points, perhaps made more surprising by the large increase observed at day 16 that correlates strongly with the enhanced GC response. We have extensively elaborated on our interpretations of how p53 might play pro-viral roles in long-term chronic infection in the updated discussion.

Fig. 4B. Annexin V is not a reliable marker of apoptosis for primary cell types. Apoptosis should be probed with reagents that directly measure caspase 3 activation.

Response: Thank you for the suggestion. We have repeated these experiments using FLICA reagent to measure active caspase 3/7 and updated the figure to include the new data.

Supplemental figure 4 legend is mislabeled.

Response: Thank you. We have corrected the legend.

MHV68-Cre has a tainted history in the field, as it is variably attenuated in wild type mice. Although supplemental Figure 1 does LD-PCR-based validation of the latent reservoir for this mutant, it is unclear how many mice/independent biological replicates were analyzed. Given the use of this suspect viral mutant in Fig. 1D, E, the observed changes need to be validated with the bla reporter MHV68 made by Tibbetts group if the authors would like to pursue two independent marking strategies.

Response: As requested and to also further close the logic loop as suggested, we employed the mLANA-Bla fusion reporter virus, as well as a newly generated M2-null counterpart, in RNA-seq analyses of infected and uninfected GC B cells. These data are included as new Fig. 6 in the revised manuscript. We have also updated the Methods section to better describe how experiments in Fig. 1d were performed.

Reviewer #2 (MHV68/ GHV) (Remarks to the Author):

The manuscript "Intrinsic p53 activation restricts gammaherpesvirus-driven germinal center B cell expansion during latent establishment" by Owens et al., reported that: p53 is activated in B cells infected with MHV68; that p53 limits MHV68 latency; and that the latent M2 protein activates a p53-dependent anti-proliferative response through a Src-mediated mechanism. Finally, they demonstrate that the EBV LMP1 protein also induces a p53 response in primary B cells. Overall, some findings are interesting; however, there are important concerns and questions that need to be solved. The lack of information makes difficult to interpret some of the results.

The authors propose that the "p53 stabilization" is not a consequence of viral infection, it

represents a cell-intrinsic reaction to the infection with MHV68. What are the mechanisms that prevent the activation of p53 as a general consequence of viral infection?.

Response: We apologize, but we do not understand what the reviewer is asking regarding “mechanisms that prevent the activation of p53 as a general consequence of infection”. However, regarding intrinsic p53 induction, the finding that p53 protein levels are increased only within infected cells (YFP+ cells, Fig. 1 and Supplementary Figs. 1 and 2) suggests that p53 induction is an intrinsic cellular response to MHV68 infection. Otherwise, we would expect p53 levels to also increase in YFP- uninfected cells within lymphoid tissues of infected mice. Increased detection of p53-related transcription when infected cells were compared to uninfected cells further supports this interpretation (Figs. 1 and 6).

Why do the authors say “p53 stabilization”? They do not evaluate p53 stability.

Response: The reviewer is correct – we did not specifically evaluate the stability of p53 protein. We have changed stabilization to induction or activation throughout the manuscript.

Acute viral replication was evaluated in lung whereas latent one was analyzed in spleen. Do the results depend on the tissue analyzed? To rule out a tissue effect, the authors should evaluate latency and acute replication in other tissues. Does acute replication of MHV68 not induce p53?

Response: After intranasal inoculation of mice, MHV68 undergoes acute replication in lung tissues before disseminating to secondary lymphoid organs and establishing latency primarily in B cells. Acute viral replication is not systemic after intranasal inoculation of immune competent mice with MHV68, is difficult to detect in organs other than the lung, and is not ongoing at the latency timepoints evaluated. However, in a previous publication we demonstrated that p53 is indeed induced, but potentially inhibited during MHV68 lytic replication (Sifford et al., JVI 2015). We therefore focused in this manuscript on the effects of p53 during viral latency. The lytic replication lung data in Supplementary Fig. 3 are included as a control to show that the absence of p53 does not lead to enhanced or uncontrolled lytic replication, since equivalent viral loads were detected for both WT and p53 KO animals.

Do the authors detect interferon production by the infected cells? If this is the case, an increase in the levels of p53 protein by the uninfected cells is expected.

Response: Thank you for this interesting comment. Like other viruses, MHV68 infection does elicit interferon production and signaling (which has been extensively studied), and p53 was previously described as a type I interferon-inducible gene. To rule out the possibility that p53 induction occurred as a consequence of the type I interferon response, we compared p53 induction in WT and IFNAR1 KO mice and found that p53 levels were identical within infected cells of both genotypes (new Supplementary Fig. 1). Consistent with our prior experiments (Fig. 1 and current Supplementary Fig. 2), p53 was not induced in the uninfected YFP- cells, which supports the conclusion that p53 induction is cell-intrinsic and interferon independent.

Why the authors do not compare the same amount of latently infected splenocytes derived from p53 ko and WT mice for being reactivated?

Response: Thank you for the question; we apologize if the experimental setup was unclear. Frequencies of reactivating cells are determined by plating equivalent numbers of latently infected cells from each infection and mouse genotype onto an indicator monolayer followed by observing cytopathic effects. Nonlinear regression analyses are performed to determine the number of reactivation competent cells based on Poisson distribution to define the statistical likelihood of 1 positive event. The reactivation efficiency, (shown in the table in Fig. 2d) is then determined by comparison to the value determined for latency evaluations using a parallel limiting-dilution PCR analysis to identify cells harboring latent viral genomes. Developed by the laboratories of Skip Virgin and Sam Speck in the 1990s, this approach is well-established in the

MHV68 field and is considered by many as the “gold standard” assays for latency and reactivation. Our results and interpretations are consistent with previous MHV68 publications.

It has previously reported that MHV68 infection triggers a DNA damage-mediated induction of p53 but when the viral cycle progresses several viral proteins inhibit the p53 pathway. How are these results explained in your model?

Response: Thank you for noting our previous work (Sifford et al., JVI 2015)! As summarized in the discussion of the current manuscript, we previously found that p53 is induced through an ATM-dependent mechanism during the lytic phase of infection, but is potentially inhibited by (at least) two viral proteins. As mentioned above, whereas our previous work focused on understanding viral interactions with p53 during the productive lytic phase of infection, the current study is focused on defining virus-p53 interactions during latency, which differs from lytic replication in that viral gene expression is restricted to a handful of “latency” genes and noncoding RNAs that facilitate maintenance of the viral genome within specific cell types without active viral replication.

It has been reported that p53 enhances IFN signaling, specifically the induction of genes containing IFN-stimulated response elements and IFN gamma secretion. Do the levels of interferon produced in response to MHV68 infection differ between p53 WT and p53 ko mice? could this have an impact on the number of cells infected in the mouse?

Response: This is an interesting possibility that could be evaluated in future studies. However, the infection phenotypes observed in p53 KO mice do not resemble those of type-I IFN non-responsive mice (increased lytic replication) or type-II IFN non-responsive mice (persistent replication and enhanced reactivation), which in our opinion suggests that p53 function is not a major contributor to IFN responses during MHV68 infection.

Many graphs do not show statistical significance.

Response: All statistically significant differences are shown. We have added text to the methods section emphasizing that results are insignificant unless otherwise noted.

Fig 1. Some graphs do not have standard deviation and it is not clear how many samples were evaluated. In addition there is no information about the statistical analysis of some graphs.

Response: The reviewer is likely referring to the qRT-PCR data in Fig. 1. For the array presentation (Fig. 1d in the revised manuscript), the data output was generated using the Qiagen profiler analysis suite. Since this analysis uses a combinatorial normalization strategy based on multiple “housekeeping” transcripts, the data are presented without error bars. However, we note that analyses were performed in technical triplicate for duplicate biological samples. We have updated the Methods description for this experiment to emphasize how the results were obtained.

Fig. 2C. There are significant differences between p53 WT and KO cells; however, it is unclear whether this lack of significance is due to the high variability. A more detailed information of the methodology and results about how the authors did the score of cytoplasmic effect would be helpful.

Response: As described for comments above, this is a well-established technique in the MHV68 pathogenesis field. The Methods section fully describes how the experiment was performed and references the original studies describing the technique.

Fig.2D. The immunohistochemical images are difficult to compare due mainly to the different B220 signal in p53WT and KO mice. The authors mention that the B220 staining was much higher in the spleens of p53 KO mice than in wt mice. Why is this happening? It has been

previously reported that M2 expression leads to a downmodulation of B220. How are these results explained in your model?

Response: We agree that there was a need for improved quality in the IHC images. We have performed new immunofluorescence analyses of WT and p53 KO mouse spleens and replaced the previous IHC images in Fig. 2. We include mock controls and unique biological replicates in Supplementary Fig. 4 to demonstrate reproducibility.

Fig.3A. Does MHV68 infection significantly increase the total number of splenocytes exclusively in p53 KO mice?

Response: No. As shown in Fig. 3a, MHV68 infection causes increases in the number of splenocytes of both WT and p53 KO mice, with this expansion being ~2-fold greater in the absence of p53.

Fig. 4a. Is there only two mock samples? More samples are needed to interpret the results, specially when there is so high deviation in the data.

Response: For this figure, the mock samples are included primarily as a reference (and gating and compensation controls, Supplementary Fig. 10) and to demonstrate that p53 deficient mice do not exhibit increased basal levels of EdU incorporation relative to WT animals. This is also consistent with equivalent numbers of cells being present in spleens of both WT and KO mice prior to infection (Fig. 3a). Importantly, the main evaluation of interest for this particular experiment is whether p53 regulates EdU incorporation after infection, and it does, particularly in GC B cells. This finding is consistent with the ~10-fold increase after infection in GL7⁺CD38^o cells detected in p53 KO spleens relative to WT mice (Fig. 3c).

Fig 4b. How many samples were evaluated? The authors did not observe statistical differences between p53WT and p53KO; however, in this case the deviation is reduced and the difference is around 10%, therefore it is hard to believe there is no significant differences.

Response: In response to comments from Reviewer 1, we repeated apoptosis experiments using a caspase 3/7 substrate. In agreement with our previous experiments using Annexin V/PI, new analyses using the caspase 3/7 substrate revealed a modest, but not significant, decrease in caspase activity after infection of p53 KO mice.

Fig. 5. Why experiment a was done at 4 days after transduction and B at 48 h after transduction?

Response: We tested a range of timepoints in preliminary experiments (24-96 hours) and found that 96 hours was best for comparisons between the different viral proteins based on reporter detection by flow cytometry (see Supplementary Fig. 11). However, 48 hours was sufficient to observe p53 induction by M2 and was more compatible with reducing cellular toxicity from drug treatments used in previous Fig. 5b (current Fig. 5c).

Fig. 6a. Is M2.Stop expression leading to an increase in the percentage of Edu+ in p53 KO cells?

Response: There is perhaps a slight increase in EdU incorporation in the control transductions of p53 KO B cells. However, the difference was not significant when compared to control transductions of WT B cells.

Figure 7 b. The authors show GFP expression at 48 h after transduction with M2.stop, M2 or LMP1: the cells look totally different and the authors state that M2 or LMP1 expression leads to a large cluster of cells. It would be helpful to show the level of expression of the different proteins at to know at what point after transduction the clusters begin to form.

Response: We apologize, but we are unsure what question the reviewer is asking us to address here. The intended point of the image is to illustrate that, relative to control transductions, both M2 and LMP1 transduced B cells (which are green due to fluorescent protein co-expression from the retroviral vector) form clusters reminiscent of activated B cells while the control transductions do not. We are unsure how a time course of cluster formation after transduction would contribute to interpretations of the experiments presented. We have included additional images in the revised manuscript showing that the other latency factors screened do not cause the clustering phenotype observed for M2 and LMP1 (Supplementary Fig. 11d), as well as additional controls showing that the latency factors are transcribed to high levels and efficiently detected for all transduced constructs (Supplementary Fig. 11c).

Analysis of the viral latency and viral proliferation in p53KO and p53 wt cells treated or not with Src inhibitors would help to evaluate the consistence of their hypothesis.

Response: This is an interesting suggestion, but it is unfortunately not feasible to treat mice with PP2 and evaluate EdU incorporation, as this would globally inhibit Src family kinase activity.

Reviewer #3 (Germinal centres/B cells/transformation) (Remarks to the Author):

In this study, Owens and Sifford, et al. employ a suite of techniques coupled with a mouse model to demonstrate in vivo that p53 is activated during latency establishment of the gammaherpesvirus MHV68. By leveraging a targeted sufficiency screen, they identify that the herpesvirus protein M2 is sufficient to stabilize p53 and to cause an anti-proliferative response. Using small molecule approaches, they uncover that this response is mediated by M2-induced activation of a Src-family kinase. The authors demonstrate functional conservation among gammaherpesviruses by expressing the EBV protein LMP1, the ortholog of M2, in mice and showing that it is sufficient to drive the same effects on p53. Overall, the paper is logical in its presentation and flow and has a number of noteworthy findings. However, I have 3 major along with a number of minor concerns outlined below. Addressing these points would strengthen the manuscript.

Major Points:

1. The authors nicely demonstrate in Figure 5a that M2 expression leads to increased p53 abundance. However, they do not quantify the levels of M2 expression compared to the other MHV68 proteins that they express in these tissues. Western blot or RT-qPCR analysis should be done to demonstrate similar levels of expression of all of the viral proteins that they are introducing, to confirm that the changes in p53 that they see are due to M2 and not a consequence of M2 being the only well-expressed protein. Also, in 5b, is there a significant difference between PP3 and PP2 treated tp53 ^{-/-} cells? Please indicate whether there is or is not, as in the PP3 and PP2 treated tp53 ^{+/+} cells on the left.

Response: Thank you for this suggestion. We have added qRT-PCR analyses demonstrating that all of the viral genes screened were transcribed at high levels within the transduced primary B cells. We have also performed new RNA-seq experiments comparing transcription profiles between GC B cells of mice infected with WT virus or M2-null MHV68. In these experiments all of the viral latency transcripts were readily detected (new Fig. 6c), however the Hallmark p53 Pathway gene set was not enriched in M2-null infections, but was for WT MHV68 (new Fig. 6e,f). These new data provide strong support for the conclusion that M2 is the main p53 agonist during MHV68 latency establishment.

Regarding Fig. 5b, a single biological replicate in the PP3 treatment group did not incorporate EdU efficiently, leading to a rather large standard error of the mean. We have added all individual data points to the revised figures to increase data transparency. We have also clarified in the Methods description of statistical analyses that all significant differences are indicated in figures.

2. The authors demonstrate that M2 promotes p53 stabilization via a Src-family kinase, though uses a large dose of the PP2 inhibitor (10 mM), which raises the possibility that there could be off-target effects. They do try to control for this with the PP3 drug, but it may have a distinct off target profile. The paper would be enhanced if the authors could show that knockdown of a specific SRC kinase produces a similar phenotype, but I realize this may be difficult to do together with M2 expression in primary cells in the timescale of the experiment. However, could it be done at a later timepoint? For instance, first express M2, then perform the shRNA control or src kinase knockdown on the proliferating tp53 +/+ vs -/- cells and analyze EdU several days later? Alternatively, can the authors demonstrate similar results with a small molecule antagonist specific for a particular SRC kinase, such as Lyn, or at least confirm with SRC inhibitors that are active at much lower doses? Or use an M2 or even LMP1 mutant that is deficient for SRC kinase activation?

Response: Thank you for these comments. The doses of all inhibitors used were based on previous publications that defined M2-related signal transduction leading to IRF4 and NFATc1 activation and IL-10 production in B cells (Rangaswamy and Speck, PLOS Path., 2014). While our findings are consistent with published results, we agree that there are potential off-target effects of the drug treatments. As the reviewer suggests, achieving successive retroviral transductions of primary B cells to knock-down specific kinases is incredibly challenging, and also not necessarily the major focus of the study as M2-dependent signal transduction was previously studied by other groups. We do agree that inclusion of non-functional M2 and LMP1 mutants would provide important specificity controls that would strengthen interpretations. We therefore generated M2 and LMP1 functional mutants based on previous publications and evaluated their capacities to activate p53 in transduced primary B cells. The results (shown in new Figs. 5b and 8b) further support the conclusion that signal transduction mediated by M2 and LMP1 leads to p53 activation in transduced cells. We have also noted in the revised text that off target effects of the drug treatments cannot be entirely ruled out.

3. The authors cover several pathways and proteins in this study. Including a final model that summarizes their key findings would be helpful for the reader to synthesize everything what they have presented.

Response: Thank you for the suggestion. We have included a general model (new Fig. 8f) as requested.

Minor Points:

1. In Figure 1a, the authors label that only 1.02% of the cells are infected and YFP+. However, this number seems low based on the rest of the data. Can the authors please explain this discrepancy?

Response: We apologize, but we are not entirely sure what the reviewer is suggesting. While the YFP+ events shown in Fig. 1a are representative, they are consistent with YFP+ percentages shown in Fig. 2a for WT mice and also agree with the corresponding LD-PCR data shown in Fig. 2b and summarized in Fig. 2d. Importantly, the results of these experiments also agree with historical published data for both types of analysis.

2. Figure 1c highlights the increases in gene expression for p53-responsive genes. While this is

evident for p53 target Cdkn1a, there is a very modest (~1 fold) change in Mdm2. This does not seem to support the argument that p53 signaling is active. Would qPCR for additional p53 target(s).

Response: We apologize for the confusion. In Fig. 1c of the original submission, we included qRT-PCR analyses for cdkn1a and mdm2 that was performed on RNA isolated from bulk splenocytes. We have removed this panel, since we show qRT-PCR data for multiple p53 responsive transcripts that are specifically upregulated in infected cells (Fig. 1d and new Fig. 6f).

3. In line 161, define what *trp53* is. Is this murine *tp53*? Also, In Figure 1e, can the authors please note how many replicates are represented by the data?

Response: Yes, Trp53 is murine p53. Replicates for previous Figure 1e (now 1d) are described in the Methods section. As noted in comments above, the normalization method used by the analysis software takes into account multiple "housekeeping" genes, which unfortunately does not allow us to show individual replicates and error bars in the figure output.

4. In Figure S3, data is shown for both 16 and 42 days post-infection. However, there is no description in the text as to the significance of these two time points. Please add some text to indicate the importance of these two time points.

Response: Thank you for this suggestion. Although not necessarily relevant for the majority of experiments presented throughout the manuscript, since we primarily focus on day 16 post-infection, we also evaluated adaptive immunity at day 42 post-infection as part of the analysis of this latency timepoint shown in Supplementary Fig. 8. Together, these findings simply show that we do not detect any overt changes in adaptive immune responses in p53-deficient mice. We have elaborated on the description of these time points in the results text.

5. In Figure S4, panel a should be panel d based on the figure legend, and panels b,c,d should be a,b,c. Please adjust the figure or the figure legend so that they match.

Response: Thank you. The figure legend has been corrected.

4. For clarity, it would help if the reactivation shown in Figure S5 could be shown as a percentage as it is shown in Figure 2. Also, Figure S5 shows that day 42 viral burden is lower in *tp53*^{-/-} mice, despite the finding that there is increased proliferation of infected cells at day 16. How can the authors reconcile this discrepancy? Is there an enhanced immune response that restricts viral infected cells between days 16 and 42, which could be an adaptive response that took time to develop? Or is there enhanced death or perhaps differentiation of infected cells after day 16?

Response: We apologize if we are misunderstanding the comment, but reactivation evaluations are not presented in the supplemental figures. As noted in new Supplementary Fig. 5, adaptive immune responses specific for MHV68 were equivalent in WT and p53 KO mice on day 42 post-infection. We speculate in the discussion as to the reason why latency is reduced at later times post-infection, however we wish to reserve any specific findings for a follow-up publication. In short, we have observed increased genomic instability after infection of p53 KO mice. We are currently wrapping up experiments for a manuscript reporting the mechanism and impact of this phenotype. As noted in response to Reviewer 1, we have also extensively revised the discussion to emphasize potential reasons for this seemingly contradictory observation.

6. Can the authors please explain how they selected the log₂ fold change cutoffs for their GSEA analysis, especially since the increase fold change cutoff is 2 but the decrease cutoff is only 1.5?

Response: We apologize for the mistake and have updated the figures to be consistent.

7. In Figure 5a, the frameshift stop mutant is not shown. However, it is discussed in the text. Can the authors please remedy this discrepancy?

Response: Although we produced and tested all of the respective frameshift-stop mutants (See Supplementary Fig. 11a,b) we only show one control (the M2.FSS) for simplicity. We have changed the figure to read “control” to hopefully make this less confusing.

8. The legend for Figure 6 refers to panels c-e, but there are only panels c-d.

Response: Thank you for pointing this out; it has been corrected.

9. In Supplementary Figure 11, why do the MYC targets go up in p53^{-/-} M2 but have the opposite trend in the LMP1 dataset?

Response: It is an interesting result that we speculate represents unique functional properties of M2 and LMP1. However, since the analysis was agnostic, we are currently unable to answer the question of “why”, but could pursue this in downstream evaluations.

10. It looks like the gating strategy for Figure 7 is shown in Supplementary Figure 10, not 11.

Response: Thank you for pointing this out; the figure legend has been corrected.

11. It would help with the clarity if the authors could refer to specific supplementary figure panels instead of the entire supplementary figure when possible.

Response: Thank you for the suggestion. We have updated the supplementary figure references as suggested.

12. In line 484, “grays” should probably be “rays.” Please check this.

Response: We are referring here to the “gray” unit of gamma radiation.

13. PMID 18242176 reported that LMP1 upregulates p53, would cite and discuss. Likewise, PMID 15330177 correlated LMP1 and p53 expression in NPC.

Response: Thank you for the suggestion. These publications were added to the discussion on lines 497-498.

14. Line 372, would consider modifying the text to indicate that people with tp53 mutations (Li Fraumeni syndrome) have just one copy of p53 mutated, as opposed to tp53 KO mice.

Response: The reviewer makes an interesting point. We were not referring to Li-Fraumeni patients per se, however we have updated the discussion (lines 443-445) to more clearly highlight that humans exhibit multiple p53 mutations and a broader spectrum of cancers than the p53 knockout mouse model, while emphasizing that the model could be used to explore whether GHV infection alters disease phenotypes when p53 is not functional. We have preliminary data indicating it does!

Response to Reviewer 1:

The revised manuscript is significantly improved, with only one comment remaining from this reviewer:

Figure 2. While the authors make a statement in the rebuttals that viral reactivation is increased in p53 deficient mice, the data in Fig. 2c do not seem to support this statement, as the SEM error bars are largely overlapping and statistically significant difference is not indicated. Similarly, data in Fig. 2b should be analyzed statistically, with additional cell dilutions analyzed by LDPCR for the p53 deficient splenocytes, as the intersection with the dotted line for this data set is derived from extrapolated curve. The authors present convincing data using marked reporter viruses; however, limiting dilution-based assays are the gold standard of the field and should complement the conclusions reached using reporter MHV68.

Response: We thank the reviewer for the opportunity to clarify our interpretations. First a comment about how the limiting-dilution analyses are performed and analyzed. As the reviewer notes, these are the historical gold-standard tests for the MHV68 field to evaluate latency (LD-PCR) and reactivation (LD-RA), and we did not deviate from protocols established in previous publications. For these assays, groups of 3-6 mice were pooled from each experimental group to reduce the impact of mouse-to-mouse variability. At least two independent infections were performed in which the virus inoculum was freshly prepared and used to infect animals on different days for each virus, condition, and time point to confirm that phenotypes are reproducible. The sample size provides about ~0.8 power to detect a difference in means of 3.1 standard deviations for each dilution which consists of either 12 or 24 data points for LD-PCR or LD-RA, respectively. The pooling of samples in each independent experimental group reduces, but does not eliminate, experimental variability. For the editor's and reviewer's evaluation, the LD-RA data from each independent infection group and the combined data from Fig. 2c follow:

Comparison of Fits	Set 1	Set 2	Combined
Null hypothesis	LogEC50 same for all data sets	LogEC50 same for all data sets	LogEC50 same for all data sets
Alternative hypothesis	LogEC50 different for each data set	LogEC50 different for each data set	LogEC50 different for each data set
P value	<0.0001	<0.0001	0.0024
Conclusion (alpha = 0.05)	Reject null hypothesis	Reject null hypothesis	Reject null hypothesis
Preferred model	LogEC50 different for each data set	LogEC50 different for each data set	LogEC50 different for each data set
F (DFn, DFd)	54.39 (1, 22)	26.43 (1, 22)	10.36 (1, 46)

The sigmoidal curves shown in these figures are best-fit models based on non-linear regression analyses performed after transforming the quantitative data points. This type of analysis is in-and-of-itself a statistical treatment that is then used to calculate a frequency of 1 positive event

(a latently infected or reactivating cell) in a given number of cells based on Poisson distribution. This is how the frequency calculations listed in Fig. 2d were defined. Extrapolated data, such as what is shown in Fig. 2b for p53^{-/-} mice represent the best-fit model, which is important when a data-set approaches the technical limits of the assay. The final dilution for the LD-PCR evaluations is approximately 30 cells per well, and unfortunately the margin for error becomes rather frustratingly unforgiving as one continues the dilution series to evaluate smaller numbers of cells. Nonetheless, as the reviewer notes, we do show in multiple, complementary approaches that MHV68 latency is enhanced in mice lacking p53. The extrapolated data from the limiting-dilution analysis presented in Fig. 2b agrees very well with the percentages shown in Fig. 2a (1 in 20 = 5% of cells, which is comparable to the ~8% determined by flow cytometry) and other quantitative data for specific cell types reported throughout the manuscript. As a further evaluation to determine whether differences between experimental groups in limiting-dilution analyses were significant, we performed Extra sum-of-squares F tests (summarized in the table above), which are appropriate for comparing non-linear regression data sets. These analyses confirmed that the non-linear regression models pass “goodness of fit” and indicate that differences between groups for the independent experiments and the combined data were indeed significant. Thus, despite the rather large error bars in the combined data set, this supports our interpretation for Fig. 2c that MHV68 reactivation is modestly enhanced in p53^{-/-} mice in a manner that correlates with increased latent infection. We have updated both Figure 2 and the Methods section to include these statistical evaluations.

RESPONSE TO REVIEWERS' COMMENTS

Reviewer #1 (Remarks to the Author):

I appreciate the author's thorough response and the technical complexity of the limiting dilution assays in the MHV68 field. Given the high SEM associated with the pooled data, data from individual experiments (limiting dilution assays pooled in the main figure) should be presented in supplemental figure.

Other than inclusion of data from individual LD-based experiments in supplemental figures, I don't have any concerns.

*Response: Thank you for your comments, we have included the individual experiment graphs in the supplemental figures (**Supplementary Fig. 3c**).*